# Different rules for binocular combination of luminance flicker in cortical and subcortical pathways

**Federico G Segala[1]\*, Aurelio Bruno[2], Joel T Martin[1], Myat T Aung[1], Alex R Wade[1,3], Daniel H Baker[1,3]**

[1]Department of Psychology, University of York, York, United Kingdom; [2]School of Psychology and Vision Sciences, University of Leicester, Leicester, United Kingdom; [3]York Biomedical Research Institute, University of York, York, United Kingdom

**Abstract** How does the human brain combine information across the eyes? It has been known for many years that cortical normalization mechanisms implement 'ocularity invariance': equalizing neural responses to spatial patterns presented either monocularly or binocularly. Here, we used a novel combination of electrophysiology, psychophysics, pupillometry, and computational modeling to ask whether this invariance also holds for flickering luminance stimuli with no spatial contrast. We find dramatic violations of ocularity invariance for these stimuli, both in the cortex and also in the subcortical pathways that govern pupil diameter. Specifically, we find substantial binocular facilitation in both pathways with the effect being strongest in the cortex. Near-linear binocular additivity (instead of ocularity invariance) was also found using a perceptual luminance matching task. Ocularity invariance is, therefore, not a ubiquitous feature of visual processing, and the brain appears to repurpose a generic normalization algorithm for different visual functions by adjusting the amount of interocular suppression.

**\*For correspondence:**
fgs502@york.ac.uk

**Competing interest:** The authors declare that no competing interests exist.

## eLife assessment

This study provides potentially **important**, new insights about the combination of information from the two eyes in humans. The data includes frequency tagging of each eye's inputs and measures reflecting both cortical (EEG) and sub-cortical processes (pupillometry). The strength of supporting evidence is **solid**, suggesting that temporal modulations are combined differently than spatial modulations, with additional differences between subcortical and cortical pathways. However, questions remain as to exactly how information is combined, how the findings relate to the extant literature and more broadly, to the interests of vision scientists at large.

## Introduction

The brain must combine information across multiple sensory inputs to derive a coherent percept of the external world. This involves a process of signal combination both within (*Baker and Wade, 2017*) and between (*Ernst and Banks, 2002*) the senses. Binocular vision is a useful test case for signal combination, as the inputs to the two eyes overlap substantially (in species with forward-facing eyes), and the neural locus is well-established (*Hubel and Wiesel, 1962*). Much of our knowledge about binocular combination derives from studies on the contrast response of the 'canonical' visual pathway, in which signals pass from the eyes to the primary visual cortex (V1), via the lateral geniculate nucleus (LGN) (*Purves et al., 2008*). However, signals are also combined across the eyes in the network of subcortical nuclei that govern pupil diameter in response to absolute light levels (*McDougal and Gamlin,*

*2008*), and much less is known about the computations that operate in these subcortical pathways. Our primary purpose here is to investigate the computations governing signal combinations in these two anatomically distinct pathways in response to luminance changes.

For pattern vision, binocular presentation confers greater sensitivity to low-contrast targets than monocular presentation. This is known as binocular summation, with summation ratios (the relative improvement under binocular presentation) at detection threshold lying between $\sqrt{2}$ and 2 (*Baker et al., 2018*; *Campbell and Green, 1965*). This advantage is lost at high stimulus contrasts, where both psychophysical performance (contrast discrimination thresholds) (*Legge, 1984*; *Meese et al., 2006*) and neural activity (*Baker and Wade, 2017*; *Moradi and Heeger, 2009*) are approximately equal for monocular and binocular presentation. Contemporary models of binocular vision (*Ding and Sperling, 2006*; *Meese et al., 2006*) advocate a process of interocular suppression that normalizes the two eyes' inputs at high contrasts and negates the binocular advantage. This is consistent with our everyday experience of 'ocularity invariance' (*Baker et al., 2007*): perceived contrast does not change when one eye is opened and closed.

The pupillary light reflex is an automatic constriction of the iris sphincter muscles in response to increases in light levels, which causes the pupil to shrink (*McDougal and Gamlin, 2008*). There is a clear binocular component to this reflex, as stimulation of one eye still causes constriction of the other eye's pupil (termed the consensual response; *Wyatt and Musselman, 1981*). Importantly, the neuroanatomical pathway involved completely bypasses the canonical cortical pathway (retina to V1), instead involving a network of subcortical nuclei, including the Pretectal Olivary nucleus, Superior Cervical ganglion, and Edinger-Westphal nucleus (*Angée et al., 2021*; *Mathôt, 2018*; *McDougal and Gamlin, 2008*; *Wang and Munoz, 2015*). To account for the consensual response, these brain regions must combine information from the left and right eyes (*ten Doesschate and Alpern, 1967*), yet the computation that achieves this is unclear. The pupil response can be modulated by periodic changes in luminance, and is temporally low-pass (*Barrionuevo et al., 2014*; *Spitschan et al., 2014*), most likely due to the mechanical limitations of the iris sphincter and dilator muscles (*Privitera and Stark, 2006*).

To investigate the binocular combination of light, we designed an experiment that allowed us to simultaneously record electrophysiological and pupillometric responses to monocular and binocular stimuli. We chose a primary flicker frequency of 2 Hz as a compromise between the low-pass pupil response (see *Barrionuevo et al., 2014*; *Spitschan et al., 2014*), and the relatively higher-pass EEG response (*Regan, 1966*). This novel paradigm allowed us to probe both cortical (using EEG) and subcortical (using a binocular eye tracker) pathways simultaneously in response to flickering light, and make quantitative comparisons between them. Periodic flicker entrains both neural (*Norcia et al., 2015*) and pupil (*Spitschan et al., 2014*) responses at the flicker frequency, enabling precise estimation of response amplitudes in the Fourier domain. Relative to the response to a monocular signal, adding a signal in the other eye can either increase the response (facilitation) or reduce it (suppression). We followed up our main experiment with an additional exploration of the effect of stimulus frequency, and a psychophysical matching experiment measuring perceived flicker intensity (i.e. temporal contrast). The results are interpreted using a hierarchical Bayesian computational model of binocular vision, and reveal that subcortical pathways implement stronger interocular suppression than the canonical cortical pathway.

## Results
### Experiment 1

The pupillometry results are summarized in *Figure 1*. The group average waveform for binocular presentation is shown in *Figure 1a*. There is a substantial pupil constriction at stimulus onset, followed by visible oscillations at the flicker frequency (2 Hz, see waveform at foot). The average Fourier spectrum is displayed in *Figure 1b*, and shows a clear spike at 2 Hz, but no evidence of a second harmonic response at 4 Hz (though see Appendix 1). These results demonstrate that our paradigm can evoke measurable steady-state pupil responses at 2 Hz.

*Figure 1c* shows contrast response functions driven by stimuli flickering only at 2 Hz. Response amplitudes increased monotonically with target contrast, confirming that our paradigm is suitable for measuring contrast-dependent differences in the pupil response (to our knowledge this is the

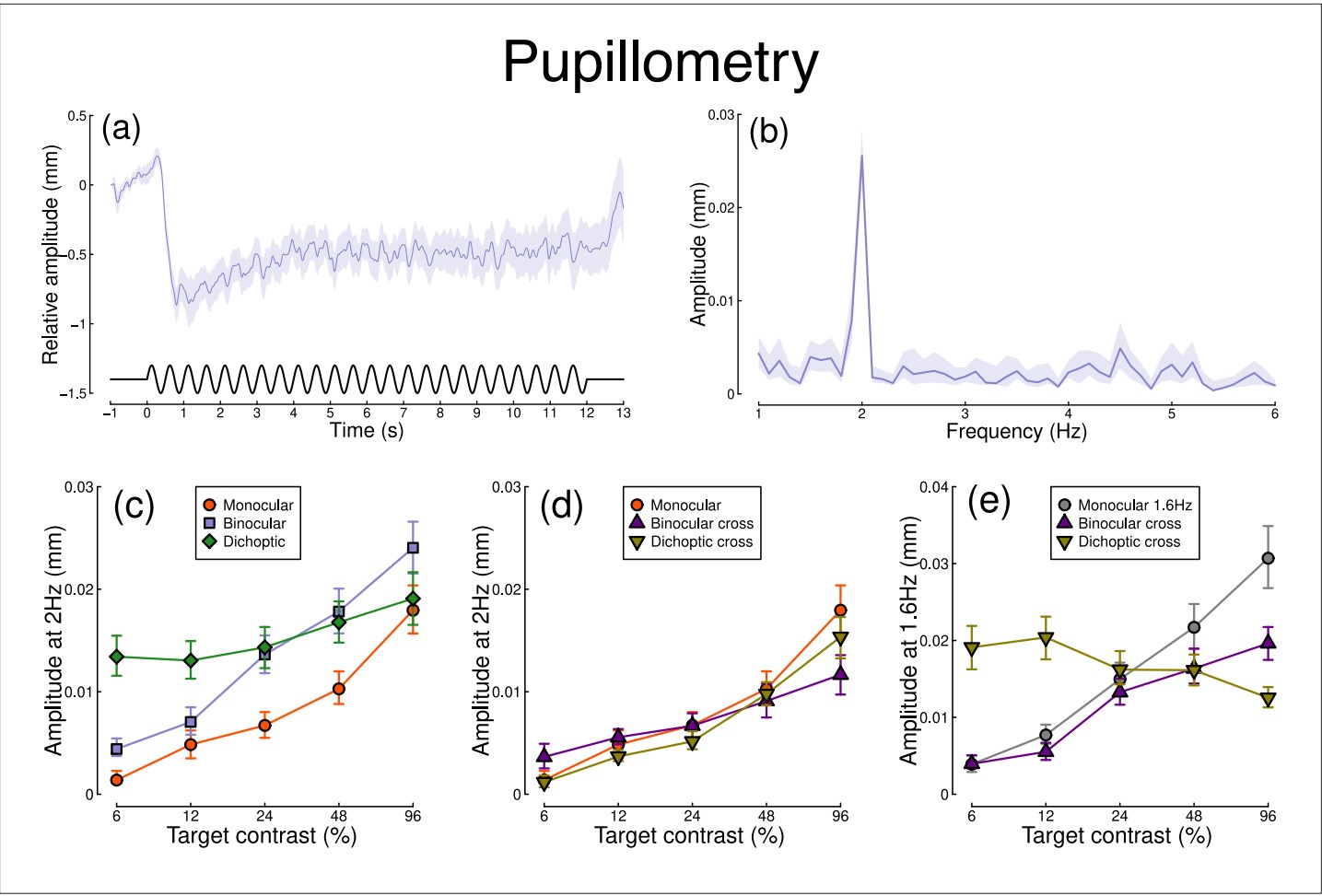

**Figure 1.** Summary of pupillometry results for N=30 participants. Panel (**a**) shows a group average waveform for binocular presentation (low pass filtered at 5 Hz), with the driving signal plotted at the foot. Negative values indicate constriction relative to baseline, and positive values indicate dilation. Panel (**b**) shows the average Fourier spectrum (absolute amplitude values). Panels (**c, d**) show contrast response functions for pupil diameter at 2 Hz for different conditions (illustrated in *Figure 8*). Panel (**e**) shows contrast response functions at 1.6 Hz for three conditions. Shaded regions and error bars indicate bootstrapped standard errors.

first time this has been demonstrated). The amplitude of the binocular condition (blue squares) is consistently greater than that of the monocular condition (red circles) across all target contrasts. A $2 \times 5$ repeated measures $ANOVA^2_{circ}$ (*Baker, 2021*) comparing these conditions revealed a significant main effect of target contrast (F(8, 580) = 16.79,  p<0.001), a significant effect of condition (F(2, 580) = 11.04,   p<0.001), and a significant interaction (F(8, 580) = 56.25, p<0.001). The dichoptic condition begins at a much higher amplitude, owing to the binocular combination of the target and high (48%) contrast mask, and then increases slightly with increasing target contrast (main effect of target contrast: F(8, 232) = 3.03, p<0.003).

In *Figure 1d*, we plot responses to monocular target stimuli flickering at 2 Hz, when the other eye viewed stimuli flickering at 1.6 Hz (the red monocular-only data are replotted from *Figure 1c* for comparison). When the 1.6 Hz component had the same contrast as the target (the binocular cross condition, shown in purple) responses were facilitated slightly at low contrasts, and suppressed at the highest target contrasts (interaction between contrast and condition: F(8, 580) = 52.94, p<0.001). When the 1.6 Hz component had a fixed contrast of 48% (the dichoptic cross condition, shown in yellow), responses were suppressed slightly across the contrast range (interaction between contrast and condition: F(8, 580) = 62.05, p<0.001).

*Figure 1e* shows responses at 1.6 Hz, for the same conditions, as well as for a condition in which a monocular stimulus flickered at 1.6 Hz (gray circles). Here, we find strong suppression in both the

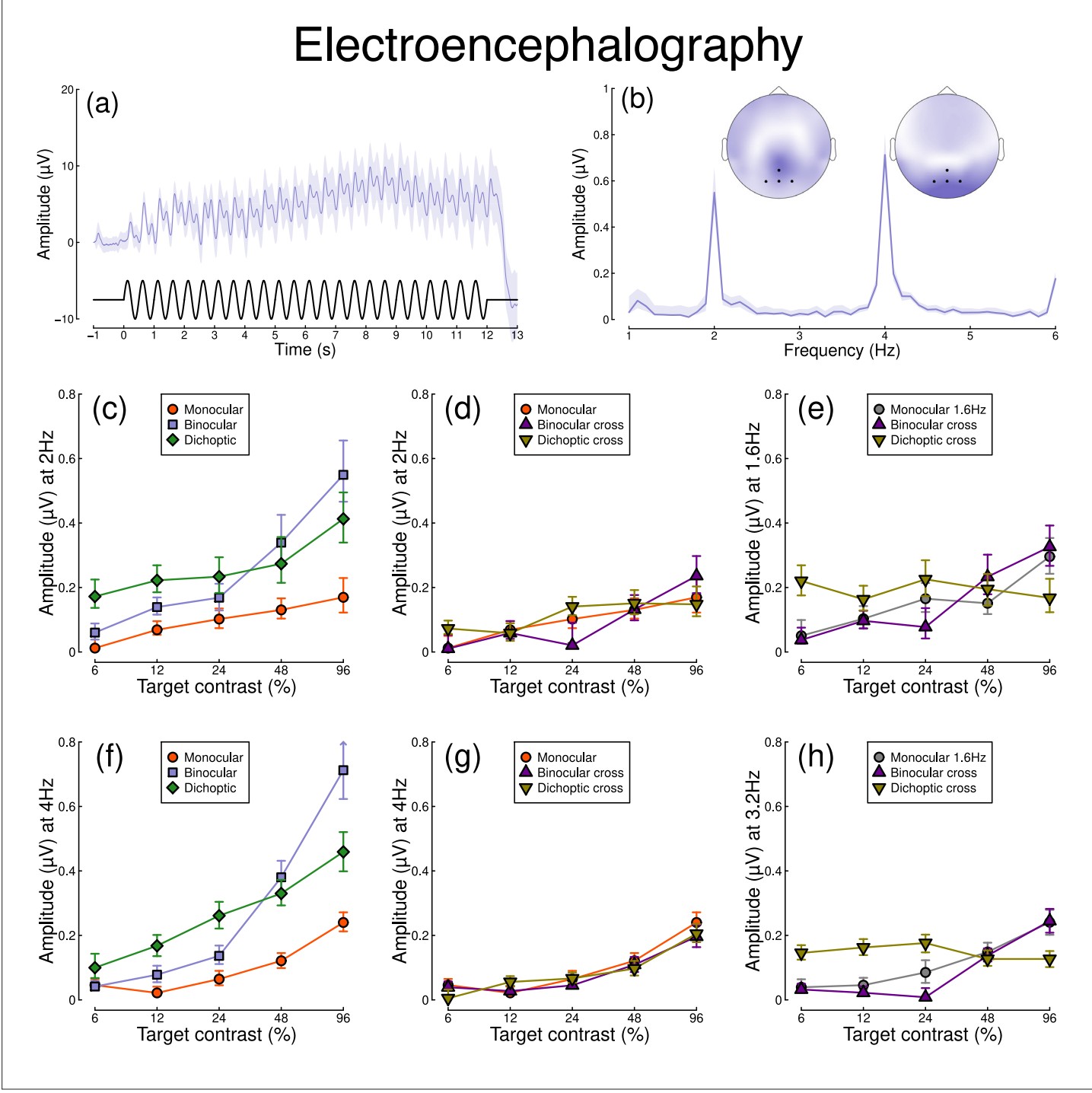

**Figure 2.** Summary of EEG results for N=30 participants. Panel (**a**) shows a group average waveform for binocular presentation (low pass filtered at 5 Hz), with the driving signal plotted at the foot. Panel (**b**) shows the average Fourier spectrum, and inset scalp distributions. Black dots on the scalp plots indicate electrodes *Oz, POz, O1,* and *O2*. Panels (**c, d**) show contrast response functions at 2 Hz for different conditions. Panel (**e**) shows contrast response functions at 1.6 Hz for three conditions. Panels (**f–h**) are in the same format but for the second harmonic response. Shaded regions and error bars indicate bootstrapped standard errors.

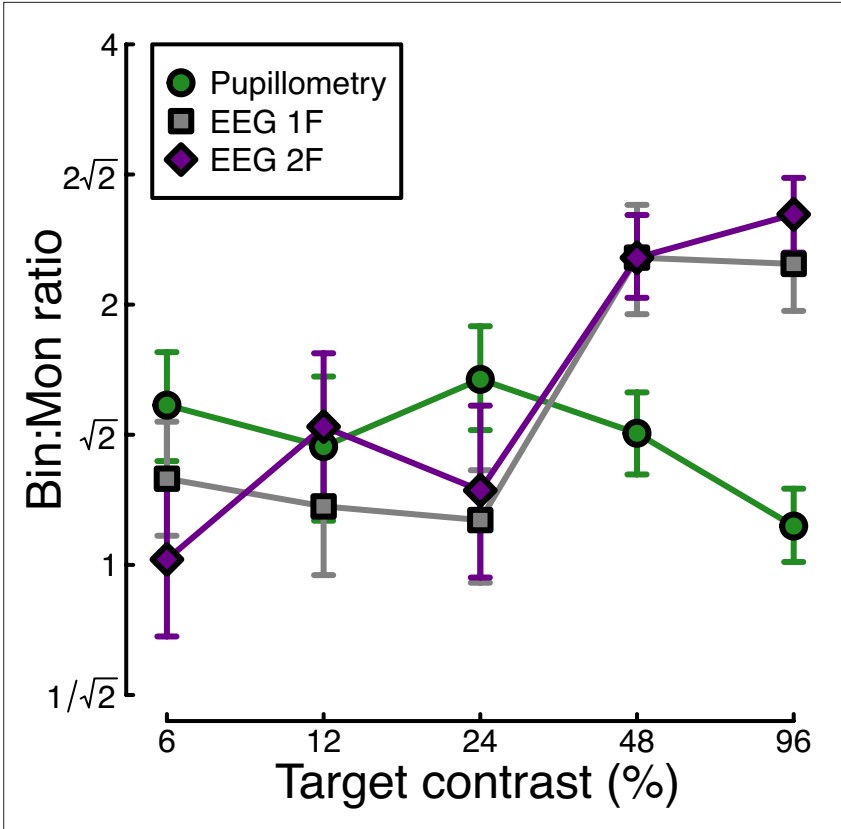

**Figure 3.** Ratio of binocular to monocular response for three data types. These were calculated by dividing the binocular response by the monocular response at each contrast level, using the data underlying **Figure 1c** and **Figure 2c, f**. Each value is the average ratio across N=30 participants, and error bars indicate bootstrapped standard errors.

binocular cross (purple triangles) and dichoptic cross (yellow triangles) conditions. In the binocular cross condition, the amplitudes are reduced relative to the monocular condition (gray circles) (interaction effect: F(8, 580) = 41.23, p<0.001). In the dichoptic cross condition, increasing the 2 Hz target contrast suppresses the response to the 1.6 Hz mask, and the function decreases (see e.g. **Busse et al., 2009**) (main effect of target contrast F(8, 232) = 17, p<0.001).

Figure 2 shows equivalent results, measured contemporaneously using EEG. **Figure 2a** shows the group average waveform for binocular presentation, and **Figure 2b** shows the Fourier spectrum for binocular presentation, both averaged across four posterior electrodes (*Oz, POz, O1,* and *O2*, marked on the inset scalp plots). Unlike for the pupillometry data, there are clear responses at both the first harmonic frequency (2 Hz), and also the second harmonic frequency (4 Hz). We therefore calculated contrast response functions at both first and second harmonic frequencies.

When stimuli in both eyes flicker at 2 Hz, the binocular responses at the first (**Figure 2c**) and second (**Figure 2f**) harmonics are substantially greater than the monocular responses, particularly at high contrasts. Analysis of variance on the complex values ($ANOVA^2_{circ}$) revealed a main effect of contrast (F(8, 580) = 4.38, p<0.001) and an interaction effect (F(8, 580) = 61.58, p<0.001), but no effect of condition (p=0.13) at the first harmonic, with a similar pattern of results obtained at the second harmonic. For the cross-frequency conditions (**Figure 2d and g**), there was no appreciable effect of adding a 1.6 Hz component on the response at 2 Hz or 4 Hz (no effect of condition, and no interaction). Similarly, there were no clear interocular interactions between frequencies in the responses at 1.6 Hz (**Figure 2e**) and 3.2 Hz (**Figure 2h**). This pattern of results suggests that the processing of temporal luminance modulations happens in a more linear way in the visual cortex (indexed by EEG), compared with subcortical pathways (indexed by pupillometry), and shows no evidence of interocular suppression.

Finally, we calculated the ratio of binocular to monocular responses across the three data types from Experiment 1. *Figure 3* shows that these ratios are approximately √2 across the low-to-intermediate contrast range for all three data types. At higher contrasts, we see ratios of 2 or higher for the EEG data, but much weaker ratios near 1 for the pupillometry data. Note that the ratios here are calculated on a per-participant basis and then averaged, rather than being the ratios of the average values shown in *Figures 1 and 2*. A $3 \times 5$ repeated measures ANOVA on the logarithmic (dB) ratios found a main effect of contrast ($F(3.08, 89.28)=4.53$, $p<0.002$), no effect of data modality ($F(2, 58) = 0.75$, $p=0.48$), but a highly significant interaction ($F(5.54, 160.67)=3.84$, $p<0.001$). All of the key results from Experiment 1 were subsequently replicated for peripheral stimulation (see Appendix 1).

## Experiment 2

The strong binocular facilitation and weak interocular suppression in the EEG data from Experiment 1 were very different from previous findings on binocular combination using steady-state EEG with grating stimuli (*Baker and Wade, 2017*). One possible explanation is that the lower temporal

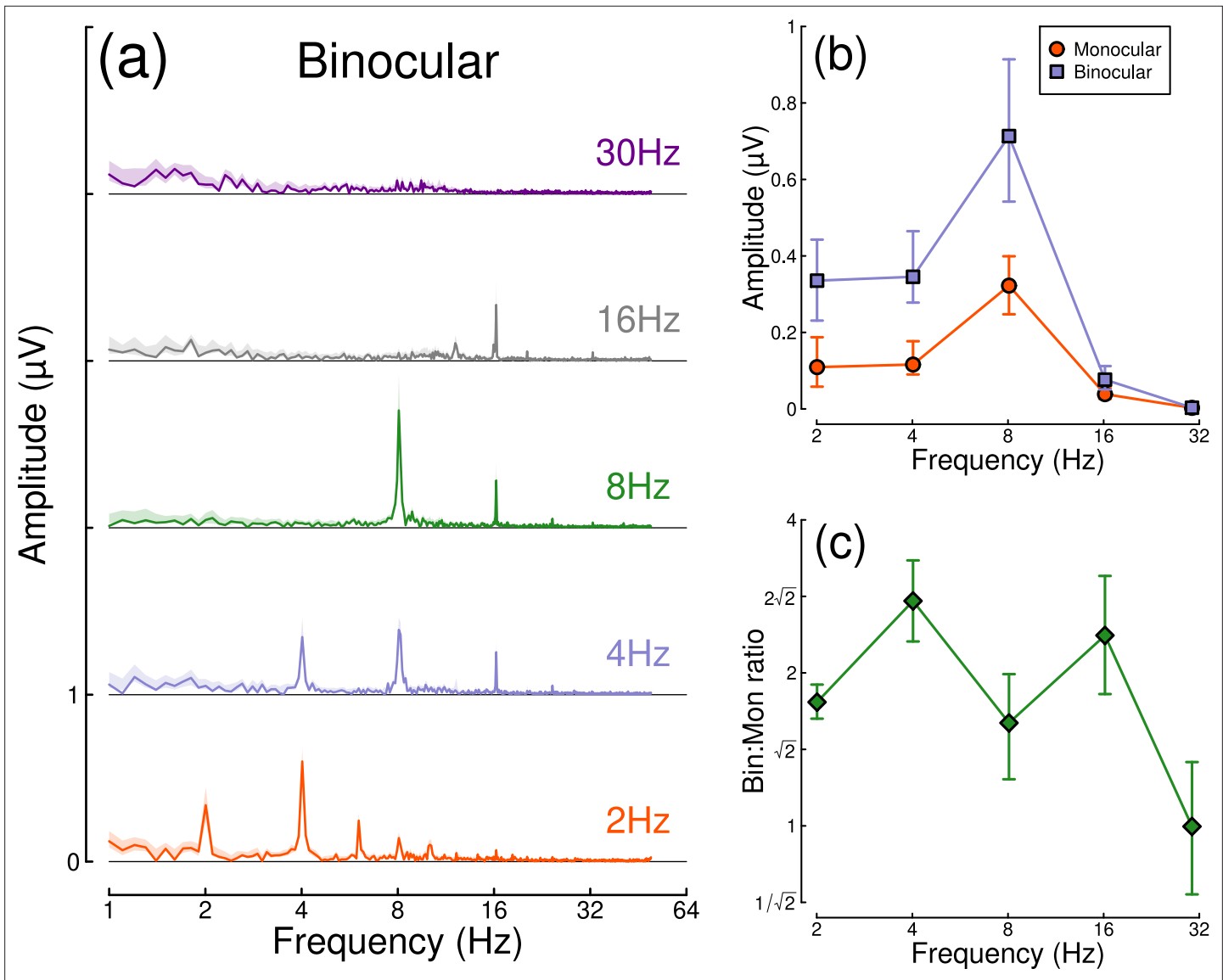

**Figure 4.** Binocular facilitation at different temporal frequencies, measured using EEG. Panel (**a**) shows Fourier spectra for responses to binocular flicker at five different frequencies (offset vertically for clarity). Panel (**b**) shows the response at each stimulation frequency for monocular (red circles) and binocular (blue squares) presentation. Panel (**c**) shows the ratio of binocular to monocular responses. Error bars and shaded regions indicate bootstrapped standard errors across N=12 participants.

frequency used here (2 Hz, vs 5 or 7 Hz in previous work) might be responsible for this difference. We, therefore, ran a second experiment to compare monocular and binocular responses at a range of temporal frequencies. Only EEG data were collected for this experiment, as the pupil response is substantially weaker above around 2 Hz (*Barrionuevo et al., 2014*; *Spitschan et al., 2014*); note that we originally chose 2 Hz because it produces measurable signals for both EEG and pupillometry, yet is unfortunately optimal for neither.

Results from the temporal frequency experiment are shown in *Figure 4*. *Figure 4a* shows the Fourier spectra for responses to binocular flicker at 5 different frequencies (2, 4, 8, 16, and 30 Hz). From 2–16 Hz, clear signals are observed at each fundamental frequency, and typically also their higher harmonics (integer multiples of the fundamental). However, at 30 Hz (upper row), the responses recorded were not demonstrably above the noise baseline. *Figure 4b* compares the monocular and binocular responses at each stimulation frequency. Here, we replicate the substantial summation effect across frequencies up to and including 16 Hz (*Figure 4c*), demonstrating that strong binocular facilitation in the EEG data of Experiment 1 cannot be attributed to our use of 2 Hz flicker.

## Experiment 3

In Experiment 1, we found evidence of stronger binocular facilitation for cortical responses to luminance flicker (measured using EEG), compared with subcortical responses (measured using pupillometry; see *Figure 3*). Since perception is dependent on cortical responses, these results provide a clear prediction for perceived contrast judgments indexed by psychophysical contrast matching paradigms (e.g. *Anstis and Ho, 1998*; *Legge and Rubin, 1981*; *Levelt, 1965*; *Quaia et al., 2018*). We therefore conducted such an experiment, in which participants judged which of two stimuli had the greater

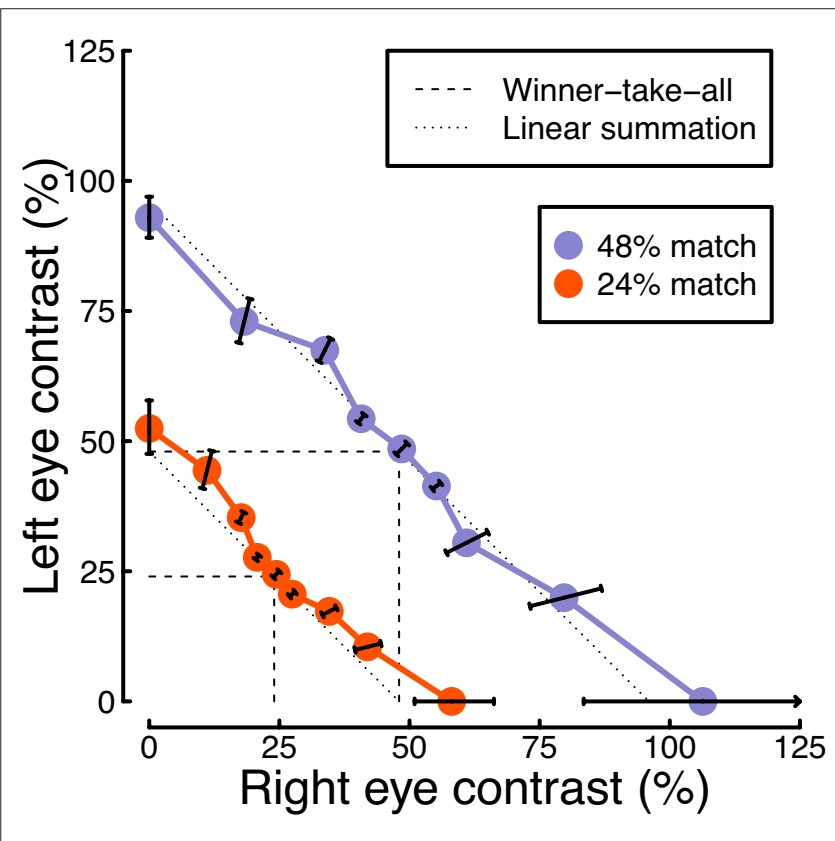

**Figure 5.** Contrast matching functions. Dotted and dashed lines are predictions of canonical summation models involving linear combination (dotted) or a winner-take-all rule (dashed). Error bars indicate the standard error across participants (N=10), and are constrained along radial lines converging at the origin. Note that, for the 48% match, the data point on the x-axis falls higher than 100% contrast. This is because the psychometric function fits for some individuals were interpolated such that the PSE fell above 100%, shifting the mean slightly above that value.

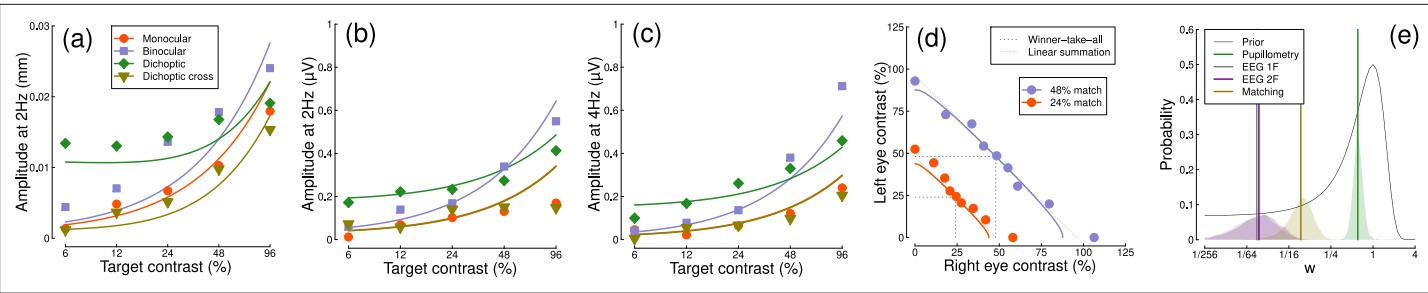

**Figure 6.** Summary of computational modeling. Panels (**a–d**) show empirical data from key conditions, replotted from earlier figures for the pupillometry (**a**), first harmonic EEG responses (**b**), second harmonic EEG responses (**c**) and contrast matching (**d**) experiments, with curves showing model behavior generated using the median group-level parameter values. Panel (**e**) shows the posterior probability distributions of the interocular suppression parameter for each of the four model fits. The pupillometry distribution (green) is centered about a substantially higher suppressive weight than for the other data types (note the logarithmic x-axis). The black curve shows the (scaled) prior distribution for the weight parameter.

perceived amplitude of flicker. On each trial, one stimulus was a matching stimulus, that had a fixed binocular flicker amplitude of either 24% or 48% (temporal) contrast. The other stimulus was a target stimulus, the contrast of which was controlled by a staircase algorithm. We tested 9 ratios of target contrast between the left and right eyes.

The results from the matching experiment are shown in *Figure 5*. Each data point indicates the contrast levels required in each eye that were perceptually equivalent to the binocular 24% (red circles) and 48% (blue circles) matching contrasts. At both matching contrasts, we see a very substantial increase in the physical contrast required for a monocular target (data points along the x- and y-axes), compared to a binocular target (points along the diagonal of x=y). For example with a 48% match, the monocular targets required contrasts close to 100%, whereas binocular targets required a contrast of around 50%. The data points between these extremes also fall close to the predictions of a linear summation model (diagonal dotted lines), and are inconsistent with a winner-takes-all (or MAX) model (dashed lines). Overall, these matching results are consistent with the approximately linear summation effects observed in the EEG data of Experiment 1 (*Figure 2c and f*).

## Computational modeling

We fitted a computational model to the data from Experiments 1 & 3 using a hierarchical Bayesian approach. The model behavior is displayed in *Figure 6a–d*, with empirical data superimposed for comparison. In general, the model captures the key characteristics of the empirical data, with group-level parameter estimates provided in *Table 1*. We were particularly interested in comparing the weight of interocular suppression across datasets. We therefore plot the posterior distributions for this parameter for all four datasets (see *Figure 6e*). The key finding is that the pupillometry results (green distribution) display a much greater weight of interocular suppression compared with the other datets (gray, purple, and yellow distributions). There is no overlap between the pupillometry distribution and any of the other three. All four distributions are also meaningfully below a weight of 1 – the value that previous work using grating stimuli would predict (*Baker and Wade, 2017*; *Meese et al., 2006*), and the peak location of our prior distribution (black curve). These results offer an explanation of the empirical data: the strong interocular suppression for the pupillometry data is consistent with the weak binocular facilitation, and measurable dichoptic masking observed using that method. The weaker suppression for the other experiments is consistent with the near-linear binocular facilitation effects, and absent dichoptic masking.

**Table 1.** Summary of median parameter values.

| Dataset | Z | n | w | Rmax |
|---|---|---|---|---|
| Pupillometry | 3.44 | 0.01 | 0.61 | 0.00023 |
| EEG 1 F | 2.62 | 0.15 | 0.02 | 0.00336 |
| EEG 2 F | 3.71 | 0.07 | 0.02 | 0.0031 |
| Matching | 0.30 | 5.10 | 0.09 | - |

## Discussion

Using a novel paradigm that combines EEG and pupillometry, we found surprising results for the binocular integration of flickering light. In the visual cortex response (indexed by EEG), the binocular combination of spatially uniform temporal luminance modulations seems to

happen approximately linearly, with no evidence of interocular suppression. Evidence for this comes from the substantial binocular facilitation effect when comparing monocular and binocular responses, and the lack of a dichoptic suppression effect when the two eyes were stimulated at different frequencies. In the subcortical pathway (indexed by pupillometry), the binocular combination is more nonlinear, with evidence of interocular suppression. This was evidenced by a weaker binocular facilitation, and stronger dichoptic suppression, relative to the EEG data. This pattern of results was confirmed by computational modeling, which showed a much greater suppressive weight for the pupillometry data compared to the EEG data. Additionally, we found that the perception of flickering light is consistent with a near-linear binocular summation process, consistent with the cortical (EEG) responses.

The results of our main experiment were unexpected for both the pupillometry and the EEG measures. Previous studies investigating binocular combination of spatial patterns (i.e. sine wave grating stimuli) are generally consistent with strong interocular suppression and weak binocular facilitation at high contrasts (*Baker and Wade, 2017*; *Meese et al., 2006*; *Moradi and Heeger, 2009*) (however, we note that facilitation as substantial as ours has been reported in previous EEG work by *Apkarian et al., 1981*). Our second experiment ruled out the possibility that these differences were due to the lower temporal frequency (2 Hz) used here. However, there is evidence of more extensive binocular facilitation for a range of other stimuli. Using scleral search coils, *Quaia et al., 2018* observed a strong binocular facilitation (or 'supersummation') in the reflexive eye movement response to rapidly moving stimuli (also known as the ocular following response). *Spitschan and Cajochen, 2019* report a similar result in archival data on melatonin suppression due to light exposure (melatonin is a hormone released by the pineal gland that regulates sleep; its production is suppressed by light exposure and can be measured from saliva assays). Work on the accommodative response indicates that binocular combination is approximately linear (*Flitcroft et al., 1992*), and can even cancel when signals are in antiphase (we did not try this configuration here). In the auditory system, interaural suppression of amplitude modulation also appears to be weak when measured using a similar steady-state paradigm (*Baker et al., 2020*). Finally, psychophysical matching experiments using static stimuli also show near-linear behavior for luminance increments (*Anstis and Ho, 1998*; *Baker et al., 2012*; *Levelt, 1965*), though not for luminance decrements (*Anstis and Ho, 1998*). Overall, this suggests that strong interocular normalization may be specific to spatial pattern vision, and not a general feature of binocular signal combination (or combination across multiple inputs in other senses).

Given the above, where does this leave our understanding of the overarching purpose of signal combination? *Baker and Wade, 2017* point out that strong suppression between channels that are subsequently summed is equivalent to a Kalman filter, which is the optimal method for combining two noisy inputs (see also *Ernst and Banks, 2002*). Functionally, interocular suppression may, therefore, act to dynamically suppress noise, rendering binocular vision more stable. This account has intuitive appeal and is consistent with other models that propose binocular combination as a means of redundancy reduction (*Li and Atick, 1994*; *May and Zhaoping, 2022*). One possibility is that optimal combination is useful for visual perception — a critical system for interacting with the local environment — and is, therefore, worth devoting the additional resource of inhibitory wiring between ocular channels. However, the other examples of binocular combination discussed above are primarily physiological responses (pupil size, eye movements, hormone release) that may benefit more from an increased signal-to-noise ratio, or otherwise be phylogenetically older than binocular pattern vision. Conceptualized another way, the brain can repurpose a generic architecture for different situational demands by adjusting parameter values (here the weight of interocular suppression) to achieve different outcomes. Our future work in this area intends to compare binocular combinations for specific photoreceptor pathways, including different cone classes, and intrinsically photoreceptive retinal ganglion cells.

Pupil size affects the total amount of light falling on the retina. It is, therefore, the case that fluctuations in pupil diameter will have a downstream effect on the signals reaching cortex. We did not incorporate such interactions into our computational model, though in principle this might be worthwhile. However, we anticipate that any such effects would be small since pupil modulations at 2 Hz are in the order of 2% of overall diameter (e.g. *Spitschan et al., 2014*). It is also the case that cortical activity can modulate pupil diameter, usually through arousal and attention mechanisms (e.g. *Bradley et al., 2008*). We think it unlikely that these temporally coarse processes would have a differential effect on e.g., monocular and binocular stimulation conditions in our experiment, and any fluctuations

during an experimental session (perhaps owing to fatigue) will be equivalent for our comparisons of interest. Therefore, we make the simplifying assumption that the pupil and perceptual pathways are effectively distinct, but hope to investigate this more directly in future neuroimaging work. Using fMRI to simultaneously image cortical and subcortical brain regions will also allow us to check that the differences we report here are not a consequence of the different measurement techniques we used (pupillometry and EEG).

Classic studies investigating the neurophysiological architecture of V1 reported that cells in cytochrome-oxidase 'blobs' (*Horton and Hubel, 1981*; *Livingstone and Hubel, 1984*) are biased towards low spatial frequencies (*Edwards et al., 1995*; *Tootell et al., 1988*), and relatively insensitive to stimulus orientation (*Horton and Hubel, 1981*; *Livingstone and Hubel, 1984*; though see *Economides et al., 2011*). As the blob regions are embedded within ocular dominance columns (*Horton and Hubel, 1981*), they are also largely monocular (*Livingstone and Hubel, 1984*; *Tychsen et al., 2004*). More recent work has reported psychophysical evidence for unoriented chromatic (*Gheiratmand et al., 2013*) and achromatic (*Meese and Baker, 2011*) mechanisms, that also appear to be monocular. Our use of luminance flicker might preferentially stimulate these mechanisms, perhaps explaining why our EEG data show little evidence of binocular interactions. Indeed, our EEG results could potentially be explained by a model involving entirely non-interacting monocular channels, with the binocular facilitation effects we find (e.g. *Figures 3 and 4*) owing to additivity of the electrophysiological response across independent monocular cells, rather than within binocular neurons. We, therefore, performed an additional analysis to investigate this possibility.

In the steady-state literature, one hallmark of a nonlinear system that pools inputs is the presence of intermodulation responses at the sums and differences of the fundamental flicker frequencies (*Baitch and Levi, 1988*; *Tsai et al., 2012*). In *Figure 7* we plot the amplitude spectra of conditions from Experiment 1 in which the two eyes were stimulated at different frequencies (2 Hz and 1.6 Hz) but at the same contrast (48%; these correspond to the binocular cross and dichoptic cross conditions in *Figures 1d, e ,, 2d and e*). *Figure 7a* reveals a strong intermodulation difference response at 0.4 Hz (red dashed line), and *Figure 7b* reveals an intermodulation sum response at 3.6 Hz (red dashed line). It seems likely that the absence of a sum response for pupillometry data, and of a difference responses for the EEG data, is a consequence of the temporal constraints of these methods. The presence of intermodulation terms is predicted by nonlinear gain control models of the type considered here (*Baker and Wade, 2017*; *Tsai et al., 2012*), and indicates that the processing of monocular flicker signals is not fully linear prior to the point at which they are combined across the eyes. Indeed, our model architecture (*Meese et al., 2006*) makes specific predictions about the location of interocular suppression - it impacts before binocular combination, consistent with results from primate physiology (*Dougherty et al., 2019*).

## Conclusions

We have demonstrated that the binocular combination of flickering light differs between cortical and subcortical pathways. Flicker was also associated with substantially weaker interocular suppression, and stronger binocular facilitation, compared to the combination of spatial luminance modulations in the visual cortex. Our computational framework for understanding signal combination permits direct comparisons between disparate experimental paradigms and data types. We anticipate that this will help elucidate the constraints the brain faces when combining different types of signals to govern perception, action, and biological function.

## Methods
### Participants

Thirty (20 females), twelve (seven females), and ten (three females) adult participants, whose ages ranged from 18 to 45, were recruited for Experiments 1, 2, and 3, respectively. All participants had normal or corrected to normal binocular vision, and gave written informed consent. Our procedures were approved by the Ethics Committee of the Department of Psychology at the University of York (identification number 792).

### Apparatus & stimuli

The stimuli were two discs of achromatic flickering light with a diameter of 3.74 degrees, presented on a black background. The same stimuli were used for all three experiments. Four dark red lines were

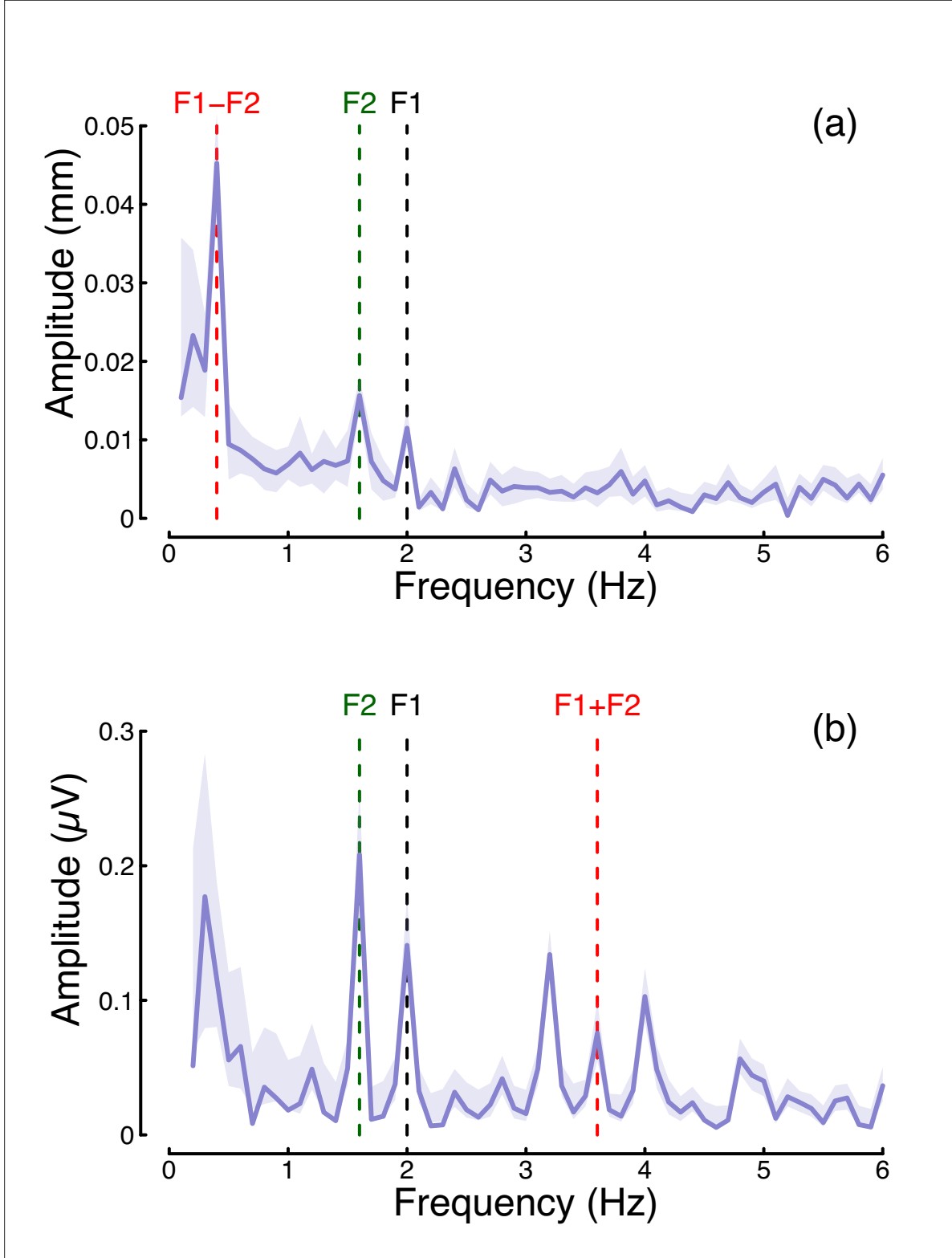

**Figure 7.** Summary of intermodulation responses in pupillometry (**a**) and EEG (**b**) data. The data are pooled across the binocular cross and dichoptic cross conditions of Experiment 1, with a target contrast of 48%. Vertical dashed lines indicate the fundamental flicker frequencies of 2 Hz (F1; black) and 1.6 Hz (F2; green), and the intermodulation difference (F1-F2=0.4 Hz) and sum (F1+F2=3.6 Hz) frequencies (red). Data are averaged across N=30 participants, and shaded regions indicate ±1 standard error.

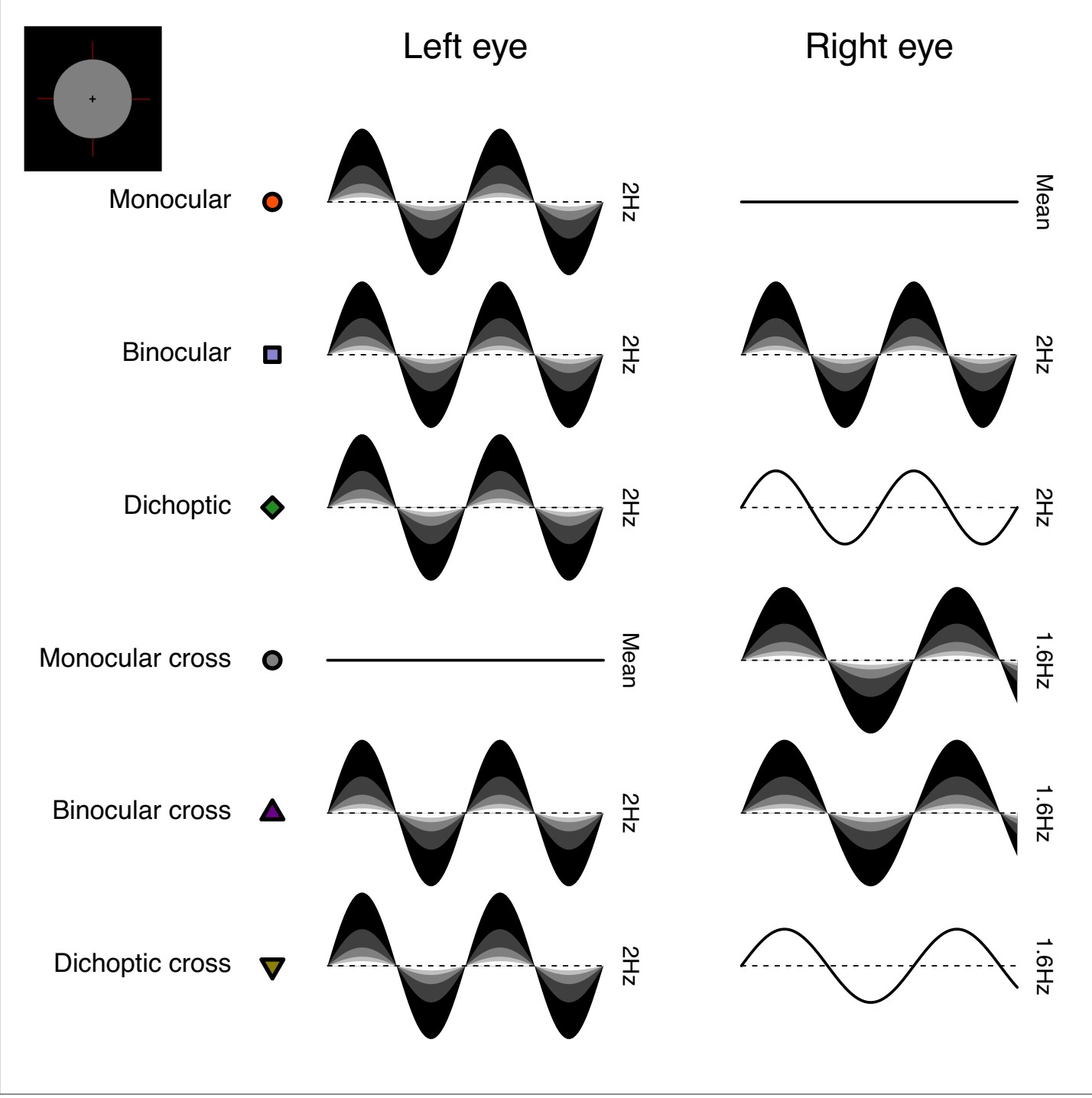

**Figure 8.** Schematic diagram illustrating the ocular arrangements, and temporal waveforms of the luminance modulations used in Experiment 1. Shaded waveforms indicate a target stimulus, that was presented at one of five contrasts on each trial (denoted by the shading levels). Unshaded waveforms indicate mask stimuli, that were presented at a fixed contrast level of 48% regardless of the target contrast. Each waveform corresponds to a 1 s period of a 12 s trial, and coloured symbols are for consistency with *Figures 1 and 2*. The icon in the upper left corner illustrates the stimulus appearance (a luminous disc against a black background). The left and right eye assignments were counterbalanced across trials in the experiment (i.e. the monocular stimulus could be shown to either eye with equal probability).

added around both discs to help with their perceptual fusion, giving the appearance of a single binocular disc (see upper left insert in *Figure 8* for an example of the fused stimulus). The discs were viewed through a four-mirror stereoscope, which used front silvered mirrors to avoid internal reflections, and meant that participants saw a single fused disc. The use of a stereoscope allowed us to modulate the stimuli in three different ocular configurations: monocular, binocular, and dichoptic. Note that during the monocular presentation of flicker, the unstimulated eye still saw the static (non-flickering) disc of mean luminance.

All stimuli had a mean luminance of 42 cd/m$^2$ and were displayed on an Iiyama VisionMaster Pro 510 display (800 × 600 pixels, 60 Hz refresh rate), which was gamma-corrected using a Minolta LS-110 photometer (Minolta Camera Co. Ltd., Japan). For experiments 1 and 2, the stimuli were presented using Psychopy (v3.0.7). For experiment 3, the stimuli were presented using Psychopy (v2022.1.1).

EEG data were collected for Experiments 1 and 2 using a 64-electrode ANT WaveGuard cap and the signals were recorded at 1 kHz using the ASA software (ANT Neuro, Netherlands). Pupillometry data were collected for Experiment 1 using a binocular Pupil Core eye-tracker (Pupil Labs GmbH, Berlin, Germany; *Kassner et al., 2014*) running at 120 Hz, and the signals were recorded with the Pupil Capture software.

## Procedure

Before each experiment, participants adjusted the angle of the stereoscope mirrors to achieve binocular fusion. This was done so that they would perceive the two discs as one fused disc when looking at the screen through the stereoscope.

### Experiment 1: simultaneous EEG and pupillometry

The experiment was conducted in a windowless room, in which the only light source was the monitor. The participants sat at 99 cm from the monitor and the total optical viewing distance (through the stereoscope) was 107 cm. The experiment was carried out in a single session lasting 45 min in total, divided into three blocks of 15 min each. In each block, there were 60 trials lasting 15 s each (12 s of stimulus presentation, with an interstimulus interval of 3 s). The participants were given no task other than to look at the fixation cross in the middle of the disc while trying to minimize their blinking during the presentation period.

We included six distinct ocular conditions, each at five temporal contrast levels (combined factorially) relative to the mean luminance: 6, 12, 24, 48, and 96%. Contrast was defined as temporal Michelson contrast; the difference between maximum and minimum luminances, scaled by the mean and expressed as a percentage. In the first three conditions, the discs flickered at 2 Hz, in either a monocular, binocular, or dichoptic arrangement (see upper rows of *Figure 8*). In the dichoptic condition, the non-target eye saw a fixed temporal contrast of 48%. The rationale for including the monocular and binocular conditions is that they permit us to measure empirically any binocular facilitation, by comparing the response amplitudes across these two conditions. The rationale for including the dichoptic condition is that it provides additional constraints to computational models, and further explores the binocular contrast-response space (see *Baker et al., 2007*).

In the remaining three conditions (termed the cross-frequency conditions) an additional flicker frequency of 1.6 Hz was introduced. We chose this frequency because it is sufficiently well-isolated from the target frequency (2 Hz) in the Fourier spectrum for 10 s trials. We repeated the monocular condition with this stimulus (one eye sees 1.6 Hz flicker, the other sees mean luminance), as well as testing in a binocular cross configuration (one eye sees each frequency at the target contrast). The rationale for the binocular cross condition is that it allows us to see the effects of suppression between the eyes without the additional complication of signal summation (which occurs when both eyes receive the same frequency), because the response of each eye can be resolved independently by frequency. Finally, in the dichoptic cross condition, one eye saw the target stimulus flickering at 2 Hz, and the other eye saw flicker at 1.6 Hz with a contrast of 48% - again this reveals the presence of suppression (by comparison with the 2 Hz monocular condition). A schematic overview of the cross-frequency conditions is shown in the lower rows of *Figure 8*. In all conditions, we counterbalanced the presentation of the target stimulus across the left and right eyes.

### Experiment 2: EEG responses across temporal frequency

This experiment used the same equipment setup as Experiment 1, except that the eye tracker was not used. Unlike the first experiment, only one contrast level was used (96%) and the discs were set to flicker at five different frequencies (2, 4, 8, 16, and 30 Hz). Only two ocular configurations, monocular and binocular, were included, with the latter having both discs flickering at the same frequency. The experiment was carried out in one session lasting 25 min in total, divided into five blocks of 5 min each. In each block, there were 20 trials in total with the same timing as for Experiment 1.

### Experiment 3: temporal contrast matching

The experiment was conducted in a darkened room with a blacked-out window. The display equipment (monitor and stereoscope) was the same as for the two previous experiments, but no EEG or pupillometry data were collected. A two-interval contrast matching procedure was used to collect data. In one interval, participants were presented with a standard fused disc that flickered at a set contrast level (either 24 or 48%), which was selected by the experimenter at the beginning of each block. In the other interval, a target disc was displayed, flickering at different contrast levels on each trial, but with a fixed interocular contrast ratio across the block. The contrast level of the target was controlled by a 1-up, 1-down staircase moving in logarithmic (dB) steps of contrast. The ratio of flicker amplitudes in the left and right eyes was varied across blocks and was set to be 0, 0.25, 0.5, 0.75, or 1 (nine distinct conditions). The standard and target discs were displayed for 1 s each, with an inter-stimulus interval of 0.5 s. After both discs had appeared on the screen, the participants had to indicate which interval they perceived as having the more intense flicker. The intervals were randomly ordered, and all discs flickered at a frequency of 2 Hz (two cycles in sine phase).

Due to its long duration (approximately 3 hr in total), the participants completed the experiment across multiple sessions initiated at their own convenience. The experiment was divided into 54 blocks (3 repetitions ×2 standard contrasts ×9 target ratios), which lasted on average 3 min each, depending on the response speed of the participant. In each block, there was a total of 50 trials. No auditory feedback was given for this subjective task.

## Data analysis

EEG data were converted from the ANT-EEProbe format to a compressed csv text file using a custom Matlab script (available at: https://github.com/bakerdh/PupillometryEEG/ copy archived at *Baker, 2023*; *Segala et al., 2023*) and components of the EEGlab toolbox (*Delorme and Makeig, 2004*). The data for each participant were then loaded into R for analysis, where a 10 s waveform for each trial at each electrode was extracted (omitting the first 2 s). The Fourier transform of each waveform was calculated, and the complex spectrum was stored in a matrix. All repetitions of each condition were then averaged for each electrode. They were then averaged across four occipital electrodes (*POz*, *Oz*, *O1*, *O2*), to obtain individual results. Finally, these were averaged across participants to obtain the group results. All averaging was performed in the complex domain and, therefore, retained the phase information (i.e. coherent averaging), and at each stage, we excluded data points with a Mahalanobis distance exceeding D = 3 from the complex-valued mean (see *Baker, 2021*). For statistical comparisons of complex-valued data, we use the $ANOVA^2_{circ}$ statistic described by *Baker, 2021*. This is a multivariate extension of ANOVA that assumes equal variance of the real and imaginary Fourier components, or equivalently, an extension of the $T^2_{circ}$ statistic of *Victor and Mast, 1991* that can compare more than two conditions.

A similar analysis pipeline was adopted for the pupillometry data. The data were converted from mp4 videos to a csv text file using the Pupil Player software (*Kassner et al., 2014*), which estimated pupil diameter for each eye on each frame using a 3D model of the eyeball. The individual data were then loaded into R for analysis, where again a 10 s waveform for each trial in each eye was extracted (excluding the first 2 s after stimulus onset). We interpolated across any dropped or missing frames to ensure regular and continuous sampling over time. The Fourier transform was calculated for each waveform, and all repetitions of each condition were pooled across the eye and then averaged. We confirmed in additional analyses that the monocular consensual pupil response was complete, justifying our pooling of data across the eyes. Finally, data were averaged across all participants to obtain the group results. Again, we used coherent averaging, and excluded outlying data points in the same way as for the EEG data. Note that previous pupillometry studies using luminance flicker have

tended to fit a single sine-wave at the fundamental frequency, rather than using Fourier analysis (e.g. *Spitschan et al., 2014*). The Fourier approach is more robust to noise at other frequencies (which can make the phase and amplitude of a fitted sine wave unstable) and has been used in some previous studies (see *Barrionuevo et al., 2014*; *Barrionuevo and Cao, 2016*). Additionally, it makes the pupillometry analysis is consistent with standard practice in steady-state EEG analysis (e.g. *Figueira et al., 2022*).

To analyze the matching data, we pooled the trial responses across all repetitions of a given condition for each participant. We then fitted a cumulative normal psychometric function to estimate the point of subjective equality at the 50% level. Thresholds were averaged across participants in logarithmic (dB) units.

For all experiments, we used a bootstrapping procedure with 1000 iterations to estimate standard errors across participants. All analysis and figure construction was conducted using a single R-script, available online, making this study fully computationally reproducible.

## Computational model and parameter estimation

To describe our data, we chose a model of binocular contrast gain control with the same general form as the first stage of the model proposed by *Meese et al., 2006*. The second gain control stage was omitted (consistent with *Baker and Wade, 2017*) to simplify the model and reduce the number of free parameters. The response of the left eye's channel is given by:

$$Resp_L = \frac{L^2}{Z + L + wR},$$ (1)

with an equivalent expression for the right eye:

$$Resp_R = \frac{R^2}{Z + R + wL}.$$ (2)

In both equations, $L$ and $R$ are the contrast signals from the left and right eyes, $Z$ is a saturation constant that shifts the contrast-response function laterally, and $w$ is the weight of suppression from the other eye.

The responses from the two eyes are then summed binocularly:

$$Resp_B = R_{max}(Resp_L + Resp_R) + n,$$ (3)

where $n$ is a noise parameter, and $R_{max}$ scales the overall response amplitude. The $R_{max}$ parameter was omitted when modeling the contrast-matching data, as it has no effect in this paradigm.

Despite being derived from the model proposed by *Meese et al., 2006*, the simplifications applied to this architecture make it very similar to other models (e.g. *Ding and Sperling, 2006*; *ten Doesschate and Alpern, 1967*; *Legge, 1984*; *Schrödinger, 1926*). In particular, we fixed the numerator exponent at 2 in our model, because otherwise, this value tends to trade off with the weight of interocular suppression (see *Baker et al., 2012*; *Kingdom and Libenson, 2015*). Our key parameter of interest is the weight of interocular suppression. Large values around w = 1 result in a very small or nonexistent binocular advantage at suprathreshold contrasts, consistent with previous work using grating stimuli (*Baker and Wade, 2017*). Low values around w = 0 produce substantial, near-linear binocular facilitation (*Baker et al., 2020*). Models from this family can handle both scalar contrast values and continuous waveforms (*Tsai et al., 2012*) or images (*Meese and Summers, 2007*) as inputs. For time-varying inputs, the calculations are performed at each time point, and the output waveform can then be analyzed using Fourier analysis in the same way as for empirical data. This means that the model can make predictions for the entire Fourier spectrum, including harmonic and intermodulation responses that arise as a consequence of nonlinearities in the model (*Baker and Wade, 2017*). However, for computational tractability, we performed fitting here using scalar contrast values.

We implemented the model within a Bayesian framework using the Stan software (*Carpenter et al., 2017*). This allowed us to estimate group-level posterior parameter distributions for the weight of interocular suppression, $w$, and the other free model parameters $R_{max}$, $Z$, and $n$. The prior distributions for all parameters were Gaussian, with means and standard deviations of 1 and 0.5 for $w$ and $R_{max}$, and 5 and 2 for $Z$ and $n$, with these values chosen based on previous literature (*Baker et al., 2012*; *Meese et al., 2006*). We sampled from a Student's t-distribution for the amplitudes in the

pupillometry and EEG experiments, and from a Bernoulli distribution for the single trial matching data. The models were fit using the individual data across all participants, independently for each dataset. We used coherent averaging to combine the data across participants, but this was not implemented in the model, so to compensate we corrected the group-level model by scaling the estimated noise parameter ($n$) by the square root of the number of participants ($n_{group} = \frac{n}{\sqrt{30}}$). We took posterior samples at over a million steps for each dataset, using a computer cluster, and retained 10% of samples for plotting.

### Preregistration, data, and code availability

We initially preregistered our main hypotheses and analysis intentions for the first experiment. We then conducted a pilot study with N=12 participants, before making some minor changes to the stimulus (we added dim red lines to aid binocular fusion). We then ran the main experiment, followed by two additional experiments that were not preregistered. The preregistration document, raw data files, and experimental and analysis code are available on the project repository: https://doi.org/10.17605/OSF.IO/TBEMA.

## Acknowledgements

Supported by Biotechnology and Biological Sciences Research Council grant BB/V007580/1 awarded to DHB and ARW, and Wellcome Trust grant 213616/Z/18/Z to AB.

## Additional information

### Funding

| Funder | Grant reference number | Author |
| --- | --- | --- |
| Biotechnology and Biological Sciences Research Council | BB/V007580/1 | Daniel H Baker<br>Alex R Wade |
| Wellcome Trust | 10.35802/213616 | Aurelio Bruno |

For the purpose of Open Access, the authors have applied a CC BY public copyright license to any Author Accepted Manuscript version arising from this submission. The funders had no role in study design, data collection and interpretation, or the decision to submit the work for publication.

### Author contributions

Federico G Segala, Data curation, Software, Formal analysis, Investigation, Visualization, Methodology, Writing – original draft, Writing – review and editing; Aurelio Bruno, Conceptualization, Supervision, Funding acquisition, Project administration, Writing – review and editing; Joel T Martin, Myat T Aung, Resources, Software, Writing – review and editing; Alex R Wade, Conceptualization, Resources, Supervision, Funding acquisition, Methodology, Project administration, Writing – review and editing; Daniel H Baker, Conceptualization, Resources, Data curation, Software, Formal analysis, Supervision, Funding acquisition, Investigation, Visualization, Methodology, Writing – original draft, Project administration, Writing – review and editing

### Author ORCIDs

Federico G Segala (iD) http://orcid.org/0000-0002-4982-8023
Aurelio Bruno (iD) http://orcid.org/0000-0002-4899-1769
Joel T Martin (iD) http://orcid.org/0000-0002-4475-3835
Alex R Wade (iD) https://orcid.org/0000-0003-4871-2747
Daniel H Baker (iD) http://orcid.org/0000-0002-0161-443X

### Ethics

All participants gave written informed consent. Our procedures were approved by the Ethics Committee of the Department of Psychology at the University of York (identification number 792).

Reviewer #1 (Public Review): https://doi.org/10.7554/eLife.87048.3.sa1
Reviewer #2 (Public Review): https://doi.org/10.7554/eLife.87048.3.sa2
Author Response https://doi.org/10.7554/eLife.87048.3.sa3

## Additional files

### Supplementary files
- MDAR checklist

### Data availability

Raw data files, and experimental and analysis code, are available on the project repository on the Open Science Framework: https://doi.org/10.17605/OSF.IO/TBEMA.

The following dataset was generated:

| Author(s) | Year | Dataset title | Dataset URL | Database and Identifier |
|---|---|---|---|---|
| Segala FG, Bruno A, Martin JT, Aung MT, Wade AR, Baker DH | 2023 | Pupillometric measures of binocular combination | https://doi.org/10.17605/OSF.IO/TBEMA | Open Science Framework, 10.17605/OSF.IO/TBEMA |

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

## Appendix 1

We conducted a conceptual replication of Experiment 1 using an alternative system of hardware and software (*Martin et al., 2023*; *Martin et al., 2022*). A pair of Spectra Tune Lab multiprimary devices (LEDmotive Technologies LLC, Barcelona, Spain) were coupled to a binocular headset using liquid light guides. The light was imaged onto a circular diffuser for each eye (field size 30 deg) with the central 8 degrees masked off using a black occluder. Therefore, the replication experiment involved peripheral stimulation, unlike the main experiment which stimulated the central ~4 degrees of the visual field. All conditions were otherwise as described for the main experiment, and we tested 12 participants in total.

The pupillometry results are shown in *Appendix 1—figure 1* and correspond closely to those from the main experiment (*Figure 1*). The ratio of binocular to monocular responses in *Appendix 1—figure 1c* is similar, and suppression is evident in *Appendix 1—figure 1d, e*. Of particular interest is the existence of a slight response at the second harmonic (4 Hz, *Appendix 1—figure 1b*), which was not present in our original data. This may be because the driving signal is stronger when stimulating the periphery (note the clear waveform in *Appendix 1—figure 1a*), or more robust to eye movements, or it might indicate additional nonlinearities not present at the fovea.

# Pupillometry

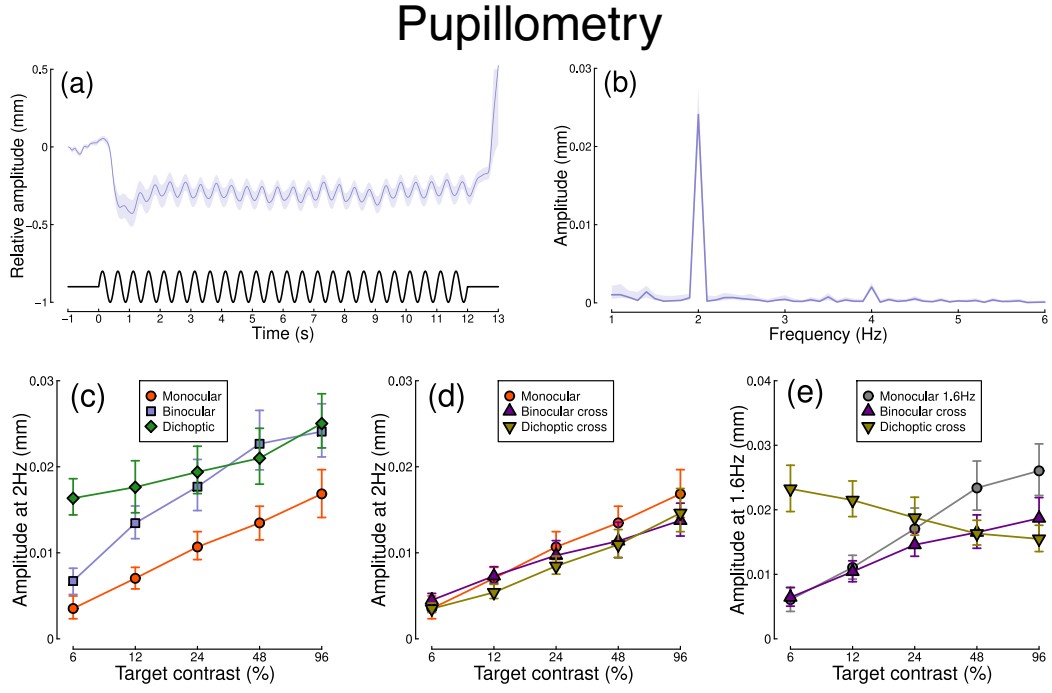

**Appendix 1—figure 1.** Summary of pupillometry results for N=12 participants, for peripheral stimulation. See *Figure 1* for a description of each panel.

The EEG results are shown in *Appendix 1—figure 2*. These still show a strong binocular facilitation effect at the highest contrast levels (*Appendix 1—figure 2c, f*), but the contrast response function is less clear at both the first and second harmonics. We suspect that this is because the cortical representation of the peripheral visual field is primarily along the calcarine sulcus, which results in some cancellation of the steady-state signal. This results in weaker signals than we obtained for foveal stimulation (represented at the occipital pole) in the main experiment (see *Figure 2*).

# Electroencephalography

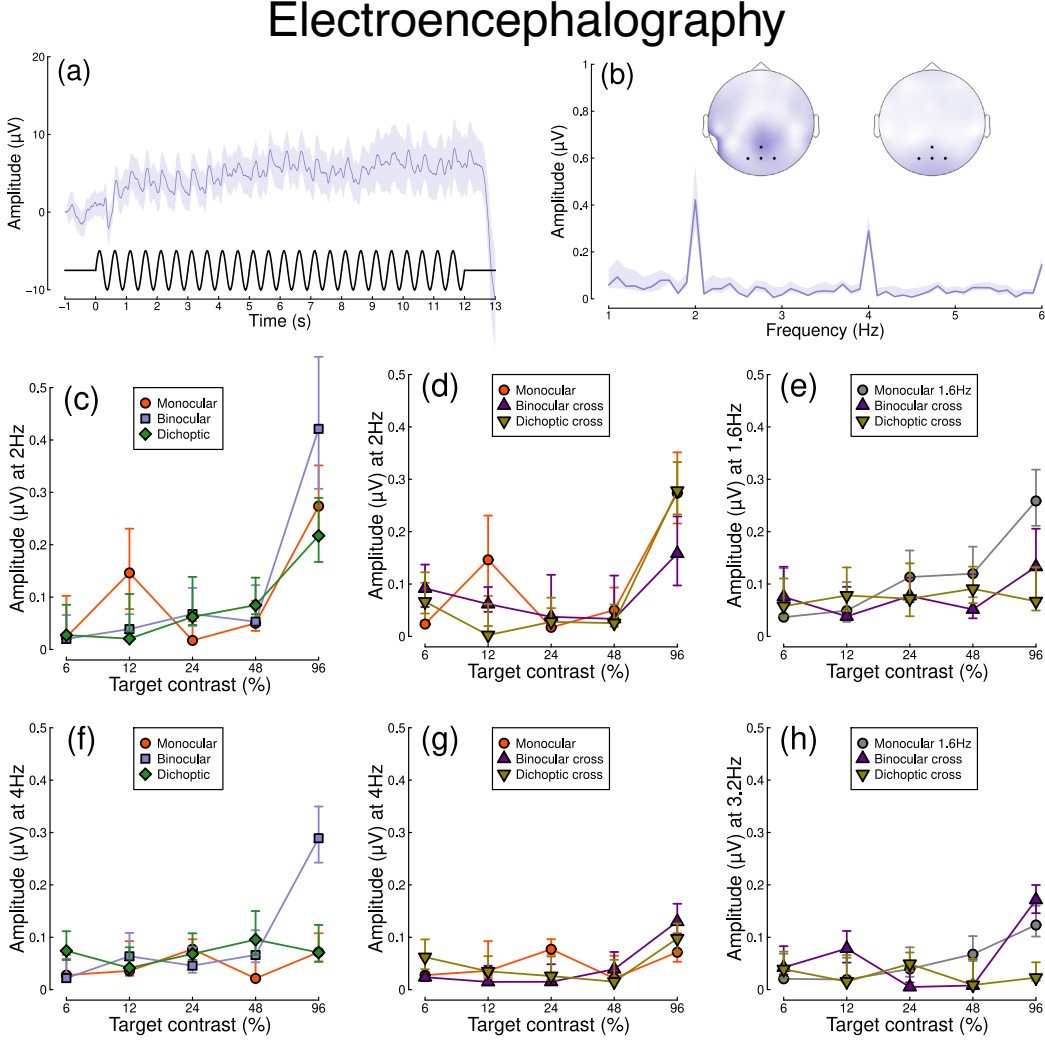

**Appendix 1—figure 2.** Summary of steady-state EEG results for N=12 participants, for peripheral stimulation. See *Figure 2* for a description of each panel.

