## [Editor Report · eLife assessment]

This study provides potentially **important**, new insights about the combination of information from the two eyes in humans. The data includes frequency tagging of each eye's inputs and measures reflecting both cortical (EEG) and sub-cortical processes (pupillometry). The strength of supporting evidence is **solid**, suggesting that temporal modulations are combined differently than spatial modulations, with additional differences between subcortical and cortical pathways. However, questions remain as to exactly how information is combined, how the findings relate to the extant literature and more broadly, to the interests of vision scientists at large.

---

## [Referee Report · Reviewer #1 (Public Review)]

In this paper, the interocular/binocular combination of temporal luminance modulations is studied. Binocular combination is of broad interest because it provides a remarkable case study of how the brain combines information from different sources. In addition, the mechanisms of binocular combination are of interest to vision scientists because they provide insight into when/where/how information from two eyes is combined.

This study focuses on how luminance flicker is combined across two eyes, extending previous work that focused mainly on spatial modulations. The results appear to show that temporal modulations are combined in different ways, with additional differences between subcortical and cortical pathways.

The manuscript has been revised to address prior reviewers' comments. It now provides more justification for the empirical choices made by the authors, and a better illustration of the methods. That said, the paper would still benefit from an expanded rationale for significance beyond this specific area. There were no substantive changes made to the abstract or introduction, and only little to the discussion.

---

## [Referee Report · Reviewer #2 (Public Review)]

Previous studies have extensively explored the rules by which patterned inputs from the two eyes are combined in visual cortex. Here the authors explore these rules for un-patterned inputs (luminance flicker) at both the level of cortex, using Steady-State Visual Evoked Potentials (SSVEPs) and at the sub-cortical level using pupillary responses. They find that the pattern of binocular combination differs between cortical and sub-cortical levels with the cortex showing less dichoptic masking and somewhat more binocular facilitation.

Importantly, the present results with flicker differ markedly from those with gratings Hou et al., 2020, J Neurosci, Baker and Wade 2017 cerebral cortex, Norcia et al, 2000 Neuroreport, Brown et al., 1999, IOVS. When SSVEP responses are measured under dichoptic conditions where each eye is driven with a unique temporal frequency, in the case of grating stimuli, the magnitude of the response in the fixed contrast eye decreases as a function of contrast in the variable contrast eye. Here the response increases by varying (small) magnitudes. The authors favor a view that cortex and perception pool binocular flicker inputs approximately linearly using cells that are largely monocular. The lack of a decrease below the monocular level when modulation strength increase is taken to indicate that previously observed normalization mechanism in pattern vision does not play a substantial role in the processing of flicker. The authors present of computational model of binocular combination that captures features of the data when fit separately to each data set. Because the model has no frequency dependence and is based on scalar quantities, it cannot make joint predictions for the multiple experimental conditions which one of its limitations.

A strength of the current work is the use of frequency-tagging of both pupil and EEG responses to measure responses for flicker stimuli at two anatomical levels of processing. Flicker responses are interesting but have been relatively neglected. The tagging approach allows one to access responses driven by each eye, even when the other eye is stimulated which is a great strength. The tagging approach can be applied at both levels of processing at the same time when stimulus frequencies are low, which is an advantage as they can be directly compared. The authors demonstrate the versatility of frequency tagging in a novel experimental design which may inspire other uses, both within the present context and others. A disadvantage of the tagging approach for studying sub-cortical dynamics via pupil responses is that it is restricted to low temporal frequencies given the temporal bandwidth of the pupil. The inclusion of a behavioral measure and a model is also a strength, but there are some limitations in the modeling (see below).

The authors suggest in the discussion that luminance flicker may preferentially drive cortical mechanisms that are largely monocular and in the results that they are approximately linear in the dichoptic cross condition (no effect of the fixed contrast stimulus in the other eye). By contrast, prior research using dichoptic dual frequency flickering stimuli has found robust intermodulation (IM) components in the VEP response spectrum (Baitch and Levi, 1988, Vision Res; Stevens et al., 1994 J Ped Ophthal Strab; France and Ver Hoeve, 1994, J Ped Ophthal Strab; Suter et al., 1996 Vis Neurosci). The presence of IM is a direct signature of binocular interaction and suggests that at least under some measurement conditions, binocular luminance combination is "essentially" non-linear, where essential implies a point-like non-linearity such as squaring of excitatory inputs. The two views are in striking contrast.

In this revised manuscript, the addition of Figure 8, which shows more complete response spectra, partially addresses this issue. However, it also raises new questions. Critically, intermodulation (IM) has to be generated at or after a point of binocular combination, as it is a mixture of the two monocular frequencies and the monocular frequencies can only mix after a point of binocular combination.

In equations 1 and 2 and in the late summation and two-stage models of Meese et al (2006), there are divisive binocular cross-links prior to a summation block. This division is a form of binocular interaction. Do equations 1 and 2 generate IM on their own with parameters used for the overall modeling? Multiplication of two inputs clearly does, as the authors indicate in their toy model. If not, then a different binocular summation rule than the one expressed in equation 3 needs to be considered to produce IM.

The discussion considers flicker processing as manifest in the EEG to be largely monocular, given the relative lack of binocular facilitation and suppression effects. And yet there is robust IM. These are difficult to reconcile as it stands. The authors suggest that their generic modeling framework can predict IM, but can it predict IM with the parameters used to fit the data, e.g. with very low values of the weight of interocular suppression and no other binocular non-linearity?

Determining whether IM can be generated by the existing non-linear elements in the model is important because previous work on dichoptic flicker IM has considered a variety of simple models of dichoptic flicker summation and has favored models involving either a non-linear combination of linear monocular inputs (Baitch and Levi, Vis Research, 1988) or a non-linear combination of rectified (non-linear) monocular inputs (Regan and Regan, Canadian J Neurol Sci, 1989). In either case, the last stage of binocular combination is non-linear, rather than linear. The authors' model is different - it has a stage of divisive binocular interaction and this "quasi-monocular" stage feeds a linear binocular combination stage.

There is a second opportunity to test the proposed model that the authors could take advantage of. In the initial review, two of the reviewers were curious about what is predicted for counter-phase inputs to the two eyes. The authors indicate that the class of models they are using could be extended to cover this case. As it turns out, this experiment has been done for dichoptic full-field flicker (Sherrington, BrJPsychiatr, 1904); van der Tweel and Estevez, Ophthalmologica, 1974; Odom and Chao, IntJNeurosci, 1995; Cavonius, QJExpPsych, 1979; Levi et al., BJO, 1982. More importantly, the predictions of several binocular combination models for anti-phase inter-ocular flicker stimulation have been tested for both the VEP and psychophysics (Odom and Chao, Int J Neurosci). Varying the relative phase of the two eyes inputs from in phase to antiphase, Odom and Chao observed that the 2nd harmonic response went to a minimum at 90 deg of interocular phase. This will happen because a 2nd order nonlinearity in the monocular path will double the phase shift of the second harmonic, putting the two eyes' 2nd harmonic response out of phase when the interocular phase is 90 deg. Summing these inputs thus leads to cancellation at 90 deg, rather than 180 deg of interocular phase. Does the authors' model predict this behavior with typical parameters used in the modeling? In the end, to account for details of both VEP and psychophysical data, Odom and Chao favored a two-path model with one path comprising non-linear monocular inputs being combined linearly and a second path combining linear monocular inputs at a non-linear binocular stage. A similar set of results and models has been developed for inter-ocular presentation of gratings (Zemon et al., PNAS, 1995).

The Odom/Chao/Zemon VEP and psychophysical data are directly relevant to the authors' work and need to be taken into account in sufficient detail so that we can judge the consistency of the proposed framework with their data and the similarities and differences in the model predictions for dichoptic flicker combination. These models are also relevant to the generation of IM, a concern raised above.

---

## [Author Response]

The following is the authors’ response to the original reviews.

**eLife assessment**
This study provides potentially important, new information about the combination of information from the two eyes in humans. The data included frequency tagging of each eye's inputs and measures reflecting both cortical (EEG) and sub-cortical processes (pupillometry). Binocular combination is of potentially general interest because it provides -in essence- a case study of how the brain combines information from different sources and through different circuits. The strength of supporting evidence appears to be solid, showing that temporal modulations are combined differently than spatial modulations, with additional differences between subcortical and cortical pathways. However, the manuscript's clarity could be improved, including by adding more convincing motivations for the approaches used.

We thank the editor and reviewers for their detailed comments and suggestions regarding our paper. We have implemented most of the suggested changes. In doing so we noticed a minor error in our analysis code that affected the functions shown in Figure 2e (previously Figure 1e), and have fixed this and rerun the modelling. Our main results and conclusions are unaffected by this change. We have also added a replication data set to the Appendix, as this bears on one of the points raised by a reviewer, and included a co-author who helped run this experiment.

**Reviewer #1 (Public Review):**
In this paper, the interocular/binocular combination of temporal luminance modulations is studied. Binocular combination is of broad interest because it provides a remarkable case study of how the brain combines information from different sources. In addition, the mechanisms of binocular combination are of interest to vision scientists because they provide insight into when/where/how information from two eyes is combined.This study focuses on how luminance flicker is combined across two eyes, extending previous work that focused mainly on spatial modulations. The results appear to show that temporal modulations are combined in different ways, with additional differences between subcortical and cortical pathways.1. Main concern: subcortical and cortical pathways are assessed in quite different ways. On the one hand, this is a strength of the study (as it relies on unique ways of interrogating each pathway). However, this is also a problem when the results from two approaches are combined - leading to a sort of attribution problem: Are the differences due to actual differences between the cortical and subcortical binocular combinations, or are they perhaps differences due to different methods. For example, the results suggest that the subcortical binocular combination is nonlinear, but it is not clear where this nonlinearity occurs. If this occurs in the final phase that controls pupillary responses, it has quite different implications.At the very least, this work should clearly discuss the limitations of using different methods to assess subcortical and cortical pathways.

The modelling asserts that the nonlinearity is primarily interocular suppression, and that this is stronger in the subcortical pathway. Moreover the suppression impacts before binocular combination. So this is quite a specific location. We now say more about this in the Discussion, and also suggest that fMRI might avoid the limits on the conclusions we can draw from different methods.

1. Adding to the previous point, the paper needs to be a better job of justifying not only the specific methods but also other details of the study (e.g., why certain parameters were chosen). To illustrate, a semi-positive example: Only page 7 explains why 2Hz modulation was used, while the methods for 2Hz modulation are described in detail on page 3. No justifications are provided for most of the other experimental choices. The paper should be expanded to better explain this area of research to non-experts. A notable strength of this paper is that it should be of interest to those not working in this particular field, but this goal is not achieved if the paper is written for a specialist audience. In particular, the introduction should be expanded to better explain this area of research, the methods should include justifications for important empirical decisions, and the discussion should make the work more accessible again (in addition to addressing the issues raised in point 1 above). The results also need more context. For example, why EEG data have overtones but pupillometry does not?

We now explain the choice of frequency in the final paragraph of the introduction as follows:

‘We chose a primary flicker frequency of 2Hz as a compromise between the low-pass pupil response (see Barrionuevo et al., 2014; Spitschan et al., 2014), and the relatively higher-pass EEG response (Regan, 1966).’

We also mention why the pupil response is low-pass:

‘The pupil response can be modulated by periodic changes in luminance, and is temporally low-pass (Barrionuevo et al., 2014; Spitschan et al. 2014), most likely due to the mechanical limitations of the iris sphincter and dilator muscles’.

**Reviewer #2 (Public Review):**
Previous studies have extensively explored the rules by which patterned inputs from the two eyes are combined in the visual cortex. Here the authors explore these rules for un-patterned inputs (luminance flicker) at both the level of the cortex, using Steady-State Visual Evoked Potentials (SSVEPs) and at the sub-cortical level using pupillary responses. They find that the pattern of binocular combination differs between cortical and sub-cortical levels with the cortex showing less dichoptic masking and somewhat more binocular facilitation.Importantly, the present results with flicker differ markedly from those with gratings (Hou et al., 2020, J Neurosci, Baker and Wade 2017 cerebral cortex, Norcia et al, 2000 Nuroreport, Brown et al., 1999, IOVS). When SSVEP responses are measured under dichoptic conditions where each eye is driven with a unique temporal frequency, in the case of grating stimuli, the magnitude of the response in the fixed contrast eye decreases as a function of contrast in the variable contrast eye. Here the response increases by varying (small) magnitudes. The authors favor a view that cortex and perception pool binocular flicker inputs approximately linearly using cells that are largely monocular. The lack of a decrease below the monocular level when modulation strength increase is taken to indicate that previously observed normalization mechanism in pattern vision does not play a substantial role in the processing of flicker. The authors present a computational model of binocular combination that captures features of the data when fit separately to each data set. Because the model has no frequency dependence and is based on scalar quantities, it cannot make joint predictions for the multiple experimental conditions which is one of its limitations.A strength of the current work is the use of frequency-tagging of both pupil and EEG responses to measure responses for flicker stimuli at two anatomical levels of processing. Flicker responses are interesting but have been relatively neglected. The tagging approach allows one to access responses driven by each eye, even when the other eye is stimulated which is a great strength. The tagging approach can be applied at both levels of processing at the same time when stimulus frequencies are low, which is an advantage as they can be directly compared. The authors demonstrate the versatility of frequency tagging in a novel experimental design which may inspire other uses, both within the present context and others. A disadvantage of the tagging approach for studying sub-cortical dynamics via pupil responses is that it is restricted to low temporal frequencies given the temporal bandwidth of the pupil. The inclusion of a behavioral measure and a model is also a strength, but there are some limitations in the modeling (see below).The authors suggest in the discussion that luminance flicker may preferentially drive cortical mechanisms that are largely monocular and in the results that they are approximately linear in the dichoptic cross condition (no effect of the fixed contrast stimulus in the other eye). By contrast, prior research using dichoptic dual frequency flickering stimuli has found robust intermodulation (IM) components in the VEP response spectrum (Baitch and Levi, 1988, Vision Res; Stevens et al., 1994 J Ped Ophthal Strab; France and Ver Hoeve, 1994, J Ped Ophthal Strab; Suter et al., 1996 Vis Neurosci). The presence of IM is a direct signature of binocular interaction and suggests that at least under some measurement conditions, binocular luminance combination is "essentially" non-linear, where essential implies a point-like non-linearity such as squaring of excitatory inputs. The two views are in striking contrast. It would thus be useful for the authors could show spectra for the dichoptic, two-frequency conditions to see if non-linear binocular IM components are present.

This is an excellent point, and one that we had not previously appreciated the importance of. We have generated a figure (Fig 8) showing the IM response in the cross frequency conditions. There is a clear response at 0.4Hz in the pupillometry data (2-1.6Hz), and at 3.6Hz in the EEG data (2+1.6Hz). We therefore agree that this shows the system is essentially nonlinear, despite the binocular combination appearing approximately linear. We now say in the Discussion:

‘In the steady-state literature, one hallmark of a nonlinear system is the presence of intermodulation responses at the sums and differences of fundamental flicker frequencies (Baitch & Levi, 1988; Tsai et al., 2012). In Figure 8 we plot the amplitude spectra of conditions from Experiment 1 in which the two eyes were stimulated at different frequencies (2Hz and 1.6Hz) but at the same contrast (48%; these correspond to the binocular cross and dichoptic cross conditions in Figures 2d,e and 3d,e). Consistent with the temporal properties of pupil responses and EEG, Figure 8a reveals a strong intermodulation difference response at 0.4Hz (red dashed line), and Figure 8b reveals an intermodulation sum response at 3.6Hz (red dashed line). The presence of these intermodulation terms is predicted by nonlinear gain control models of the type considered here (Baker and Wade, 2017; Tsai et al., 2012), and indicates that the processing of monocular flicker signals is not fully linear prior to the point at which they are combined across the eyes.’

If the IM components are indeed absent, then there is a question of the generality of the conclusions, given that several previous studies have found them with dichoptic flicker. The previous studies differ from the authors' in terms of larger stimuli and in their use of higher temporal frequencies (e.g. 18/20 Hz, 17/21 Hz, 6/8 Hz). Either retinal area stimulated (periphery vs central field) or stimulus frequency (high vs low) could affect the results and thus the conclusions about the nature of dichoptic flicker processing in cortex. It would be interesting to sort this out as it may point the research in new directions.

This is a great suggestion about retinal area. As chance would have it, we had already collected a replication data set where we stimulated the periphery, and we now include a summary of this data set as an Appendix. In general the results are similar, though we obtain a measurable (though still small) second harmonic response in the pupillometry data with this configuration, which is a further indication of nonlinear processing.

Whether these components are present or absent is of interest in terms of the authors' computational model of binocular combination. It appears that the present model is based on scalar magnitudes, rather than vectors as in Baker and Wade (2017), so it would be silent on this point. The final summation of the separate eye inputs is linear in the model. In the first stage of the model, each eye's input is divided by a weighted input from the other eye. If we take this input as inhibitory, then IM would not emerge from this stage either.

We have performed the modelling using scalar values here for simplicity and transparency, and to make the fitting process computationally feasible (it took several days even done this way). This type of model is quite capable of processing sine waves as inputs, and producing a complex output waveform which is Fourier transformed and then analysed in the same way as the experimental data (see e.g. Tsai, Wade & Norcia, 2012, J Neurosci; Baker & Wade, 2017, Cereb Cortex). However our primary aim here was to fit the model, and make inferences about the parameter values, rather than to use a specific set of parameter values to make predictions.We now say more about this family of models and how they can be applied in the methods section:

“Models from this family can handle both scalar contrast values and continuous waveforms (Tsai et al., 2012) or images (Meese and Summers, 2007) as inputs. For time-varying inputs, the calculations are performed at each time point, and the output waveform can then be analysed using Fourier analysis in the same way as for empirical data.This means that the model can make predictions for the entire Fourier spectrum, including harmonic and intermodulation responses that arise as a consequence of nonlinearities in the model (Baker and Wade, 2017). However for computational tractability, we performed fitting here using scalar contrast values.”

As a side point, there are quite a lot of ways to produce intermodulation terms, meaning they are not as diagnostic as one might suppose. We demonstrate this in Author response image 1, which shows the Fourier spectra produced by a toy model that multiplies its two inputs together (for an interactive python notebook that allows various nonlinearities to be explored, see here). Intermodulation terms also arise when two inputs of different frequencies are summed, followed by exponentiation. So it would be possible to have an entirely linear binocular summation process, followed by squaring, and have this generate IM terms (not that we think this is necessarily what is happening in our experiments).

Related to the model: One of the more striking results is the substantial difference between the dichoptic and dichoptic-cross conditions. They differ in that the latter has two different frequencies in the two eyes while the former has the same frequency in each eye. As it stands, if fit jointly on the two conditions, the model would make the same prediction for the dichoptic and dichoptic-cross conditions. It would also make the same prediction whether the two eyes were in-phase temporally or in anti-phase temporally. There is no frequency/phase-dependence in the model to explain differences in these cases or to potentially explain different patterns at the different VEP response harmonics. The model also fits independently to each data set which weakens its generality. An interpretation outside of the model framework would thus be helpful for the specific case of differences between the dichoptic and dichoptic-cross conditions.

As mentioned above, the limitations the reviewer highlights are features of the specific implementation, rather than the model architecture in general. Furthermore, although this particular implementation of the model does not have separate channels for different phases, these can be added (see e.g. Georgeson et al., 2016, Vis Res, for an example in the spatial domain). In future work we intend to explore the phase relationship of flicker, but do not have space to do this here.

Prior work has defined several regimes of binocular summation in the VEP (Apkarian et al.,1981 EEG Journal). It would be useful for the authors to relate the use of their terms "facilitation" and "suppression" to these regimes and to justify/clarify differences in usage, when present. Experiment 1, Fig. 3 shows cases where the binocular response is more than twice the monocular response. Here the interpretation is clear: the responses are super-additive and would be classed as involving facilitation in the Apkarian et al framework. In the Apkarian et al framework, a ratio of 2 indicates independence/linearity. Ratios between 1 and 2 indicate sub-additivity and are diagnostic of the presence of binocular interaction but are noted by them to be difficult to interpret mechanistically. This should be discussed. A ratio of <1 indicates frank suppression which is not observed here with flicker.

Operationally, we use facilitation to mean an increase in response relative to a monocular baseline, and suppression to mean a decrease in response. We now state this explicitly in the Introduction. Facilitation greater than a factor of 2 indicates some form of super-additive summation. In the context of the model, we also use the term suppression to indicate divisive suppression between channels, however this feature does not always result in empirical suppression (it depends on the condition, and the inhibitory weight). We think that interpretation of results such as these is greatly aided by the use of a computational modelling framework, which is why we take this approach here. The broad applicability of the model we use in the domain of spatial contrast lends it credibility for our stimuli here.

Can the model explore the full range of binocular/monocular ratios in the Apkarian et al framework? I believe much of the data lies in the "partial summation" regime of Apkarian et al and that the model is mainly exploring this regime and is a way of quantifying varying degrees of partial summation.

Yes, in principle the model can produce the full range of behaviours. When the weight of suppression is 1, binocular and monocular responses are equal. When the weight is zero, the model produces linear summation. When the weight is greater than 1, suppression occurs. It is also possible to produce super-additive summation effects, most straightforwardly by changing the model exponents. However this was not required for our data here, and so we kept these parameters fixed. We agree that the model is a good way to unify the results across disparate experimental paradigms, and that is our main intention with Figure 7i.

**Reviewer #3 (Public Review):**
This manuscript describes interesting experiments on how information from the two eyes is combined in cortical areas, sub-cortical areas, and perception. The experimental techniques are strong and the results are potentially quite interesting. But the manuscript is poorly written and tries to do too much in too little space. I had a lot of difficulty understanding the various experimental conditions, the complicated results, and the interpretations of those results. I think this is an interesting and useful project so I hope the authors will put in the time to revise the manuscript so that regular readers like myself can better understand what it all means.Now for my concerns and suggestions:The experimental conditions are novel and complicated, so readers will not readily grasp what the various conditions are and why they were chosen. For example, in one condition different flicker frequencies were presented to the two eyes (2Hz to one and 1.6Hz to the other) with the flicker amplitude fixed in the eye presented to the lower frequency and the flicker amplitude varied in the eye presented to the higher frequency. This is just one of several conditions that the reader has to understand in order to follow the experimental design. I have a few suggestions to make it easier to follow. First, create a figure showing graphically the various conditions. Second, come up with better names for the various conditions and use those names in clear labels in the data figures and in the appropriate captions. Third, combine the specific methods and results sections for each experiment so that one will have just gone through the relevant methods before moving forward into the results. The authors can keep a general methods section separate, but only for the methods that are general to the whole set of experiments.

We have created a new figure (now Fig 1) that illustrates the conditions from Experiment 1, and is referenced throughout the paper. We have kept the names constant, as they are rooted in a substantial existing literature, and it will be confusing to readers familiar with that work if we diverge from these conventions. We did consider separating out the methods section, but feel it helps the flow of the results section to keep it as a single section.

I wondered why the authors chose the temporal frequencies they did. Barrionuevo et al (2014) showed that the human pupil response is greatest at 1Hz and is nearly a log unit lower at 2Hz (i.e., the change in diameter is nearly a log unit lower; the change in area is nearly 2 log units lower). So why did the authors choose 2Hz for their primary frequency? And why did the authors choose 1.6Hz which is quite close to 2Hz for their off frequency? The rationale behind these important decisions should be made explicit.

We now explain this in the Introduction as follows:

‘We chose a primary flicker frequency of 2Hz as a compromise between the low-pass pupil response (see Barrionuevo et al., 2014; Spitschan et al., 2014), and the relatively higher-pass EEG response (Regan, 1966).’

It is a compromise frequency that is not optimal for either modality, but generates a measurable signal for both. The choice of 1.6 Hz was for similar reasons - for a 10-second trial it is four frequency bins away from the primary frequency, so can be unambiguously isolated in the spectrum.

By the way, I wondered if we know what happens when you present the same flicker frequencies to the two eyes but in counter-phase. The average luminance seen binocularly would always be the same, so if the pupil system is linear, there should be no pupil response to this stimulus. An experiment like this has been done by Flitcroft et al (1992) on accommodation where the two eyes are presented stimuli moving oppositely in optical distance and indeed there was no accommodative response, which strongly suggests linearity.

We have not tried this yet, but it’s on our to-do list for future work. The accommodation work is very interesting, and we now cite it in the manuscript as follows:

‘Work on the accommodative response indicates that binocular combination there is approximately linear (Flitcroft et al. 1992), and can even cancel when signals are in antiphase (we did not try this configuration here).’

Figures 1 and 2 are important figures because they show the pupil and EEG results, respectively. But it's really hard to get your head around what's being shown in the lower row of each figure. The labeling for the conditions is one problem. You have to remember how "binocular" in panel c differs from "binocular cross" in panel d. And how "monocular" in panel d is different than "monocular 1.6Hz" in panel e. Additionally, the colors of the data symbols are not very distinct so it makes it hard to determine which one is which condition. These results are interesting. But they are difficult to digest.

We hope that the new Figure 1 outlining the conditions has helped with interpretation here.

The authors make a strong claim that they have found substantial differences in binocular interaction between cortical and sub-cortical circuits. But when I look at Figures 1 and 2, which are meant to convey this conclusion, I'm struck by how similar the results are. If the authors want to continue to make their claim, they need to spend more time making the case.

Indeed, it is hard to make direct comparisons across figures - this is why Figure 4 plots the ratio of binocular to monocular conditions, and shows a clear divergence between the EEG and pupillometry results at high contrasts.

Figure 5 is thankfully easy to understand and shows a very clear result. These perceptual results deviate dramatically from the essentially winner-take-all results for spatial sinewaves shown by Legge & Rubin (1981); whom they should cite by the way. Thus, very interestingly the binocular combination of temporal variation is quite different than the binocular combination of spatial variation. Can the pupil and EEG results also be plotted in the fashion of Figure 5? You'd pick a criterion pupil (or EEG) change and use it to make such plots.

We now cite Legge & Rubin. We see what you mean about plotting the EEG and pupillometry results in the same coordinates as the matching data, but we don’t think this is especially informative as we would end up only with data points along the axes and diagonal of the plot, without the points at other angles. This is a consequence of how the experiments were conducted.

My main suggestion is that the authors need to devote more space to explaining what they've done, what they've found, and how they interpret the data. I suggest therefore that they drop the computational model altogether so that they can concentrate on the experiments. The model could be presented in a future paper.

We feel that the model is central to the understanding and interpretation of our results, and have retained it in the revised version of the paper.

**Reviewer #2 (Recommendations For The Authors):**
I found the terms for the stimulus conditions confusing. I think a simple schematic diagram of the conditions would help the reader.

Now added (the new Fig 1).

In reporting the binocular to monocular ratio, please clarify whether the monocular data was from one eye alone (and how that eye was chosen) or from both eyes and then averaged, or something else. It would be useful to plot the results from the dichoptic condition in this form, as well.

These were averaged across both eyes. We now say in the Methods section:

‘We confirmed in additional analyses that the monocular consensual pupil response was complete, justifying our pooling of data across the eyes.’

Also, clarify whether the term facilitation is used as above throughout (facilitation being > 2 times monocular response under binocular condition) or if a different criterion is being used. If we take facilitation to mean a ratio > 2, then facilitation depends on temporal frequency in Figure 4.

We now explain our use of these terms in the final paragraph of the Introduction:

‘Relative to the response to a monocular signal, adding a signal in the other eye can either increase the response (facilitation) or reduce it (suppression).’

The magnitude of explicit facilitation attained is interesting, but not without precedent. Ratios of binocular to mean monocular > 2, have been reported previously and values of summation depend strongly on the stimulus used (see for example Apkarian et al., EEG Journal, 1981, Nicol et al., Doc Ophthal, 2011).

We now mention this in the Discussion as follows:

‘(however we note that facilitation as substantial as ours has been reported in previous EEG work by Apkarian et al. (1981))’

In Experiment 3, the authors say that the psychophysical matching results are consistent with the approximately linear summation effects observed in the EEG data of Experiment 1. In describing Fig. 3, the claim is that the EEG is non-linear, e.g. super-additive - at least at high contrasts. Please reconcile these statements.

We think that the ‘superadditive’ effects are close enough to linear that we don’t want to make too much of a big deal about them - this could be measurement error, for example. So we use terms such as near-linear, or approximately linear, when referring to them throughout.

**Reviewer #3 (Recommendations For The Authors):**
Let me make some more specific comments using a page/paragraph/line format to indicate where in the text they're relevant.1/2 (middle)/3 from end. "In addition" seems out of place here.

Removed.

1/3/4. By "intensities" do you mean "contrasts"?

Fixed.

1/3/last. "... eyes'...".

Fixed.

2/5/3. By "one binocular disc", you mean into "one perceptually fused disc".

Rewritten as: ‘to help with their perceptual fusion, giving the appearance of a single binocular disc’

3/1/1. "calibrated" seems like the wrong word here. I think you're just changing the vergence angle to enable fusion, right?

Now rewritten as: ‘Before each experiment, participants adjusted the angle of the stereoscope mirrors to achieve binocular fusion’

3/1/1. "adjusting the angles...". And didn't changing the mirror angles affect the shapes of the discs in the retinal images?

Perhaps very slightly, but this is well within the tolerance of the visual system to compensate for in the fused image, especially for such high contrast edges.

3/3/5. "fixed contrast" is confusing here because it's still a flickering stimulus if I follow the text here. Reword.

Now ‘fixed temporal contrast’

3/4/1. It would be clearer to say "pupil tracker" rather than "eye tracker" because you're not really doing eye tracking.

True, but the device is a commercial eye tracker, so this is the appropriate term regardless of what we are using it for.

3/5/6. I'm getting lost here. "varying contrast levels" applies to the dichoptic stimulus, right?

Yes, now reworded as ‘In the other interval, a target disc was displayed, flickering at different contrast levels on each trial, but with a fixed interocular contrast ratio across the block.’

3/5/7. Understanding the "ratio of flicker amplitudes" is key to understanding what's going on here. More explanation would be helpful.

Addressed in the above point.

4/3/near end. Provide some explanation about why the Fourier approach is more robust to noise.

Added ‘(which can make the phase and amplitude of a fitted sine wave unstable)’

Figure 1. In panel a, explain what the numbers on the ordinate mean. What's zero, for example? Which direction is dilation? Same question for panel b. It's interesting in panel c that the response in one eye to 2Hz increases when the other eye sees 1.6Hz. Would be good to point that out in the text.

Good idea about panel (a) - we have changed the y-axis to ‘Relative amplitude’ for clarity, and now note in the figure caption that ‘Negative values indicate constriction relative to baseline, and positive values indicate dilation.’ Panel (b) is absolute amplitude, so is unsigned. Panel (c) only contains 2Hz conditions, but there is some dichoptic suppression across the two frequencies in panels (d,e) - we now cover this in the text and include statistics.

6/2/1. Make clear in the text that Figure 1c shows contrast response functions for the pupil.

Now noted in the caption.

Figure 3. I'm lost here. I feel like I should be able to construct this figure from Figures 1 and 2, but don't know how. More explanation is needed at least in the caption.

Done. The caption now reads:

‘Ratio of binocular to monocular response for three data types. These were calculated by dividing the binocular response by the monocular response at each contrast level, using the data underlying Figures 2c, 3c and 3f. Each value is the average ratio across N=30 participants, and error bars indicate bootstrapped standard errors.’

9/1/1-2. I didn't find the evidence supporting this statement compelling.

We now point the reader to Figure 4 as a reminder of the evidence for this difference.

9/1/6-9. You said this. But this kind of problem can be fixed by moving the methods sections as I suggested above.

As mentioned, we feel that the results section flows better with the current structure.

Figure 4. Make clear that this is EEG data.

Now added to caption.

Figure 5 caption. Infinite exponent in what equation?

Now clarified as: ‘models involving linear combination (dotted) or a winner-take-all rule (dashed)’

Figure 6. I hope this gets dropped. No one will understand how the model predictions were derived. And those who look at the data and model predictions will surely note (as the authors do) that they are rather different from one another.

As noted above, we feel that the model is central to the paper and have retained this figure. We have also worked out how to correct the noise parameter in the model for the number of participants included in the coherent averaging, which fixes the discrepancy at low contrasts. The correspondence between the data and model in is now very good, and we have plotted the data points and curves in the same panels, which makes the figure less busy.

12/1. Make clear in this paragraph that "visual cortex" is referring to EEG and perception results and that "subcortical" is referring to pupil. Explain clearly what "linear" would be and what the evidence for "non-linear" is.

Good suggestion, we have added qualifiers linking to both methods. Also tidied up the language to make it clearer that we are talking about binocular combination specifically in terms of linearity, and spelled out the evidence for each point.

12/2/6-9. Explain the Quaia et al results enough for the reader to know what reflexive eye movements were studied and how.

We now specify that these eye movements are also known as the ‘ocular following response’ and were measured using scleral search coils.

12/2/9-10. Same for Spitchan and Cajochen: more explanation.

Added:

“(melatonin is a hormone released by the pineal gland that regulates sleep; its production is suppressed by light exposure and can be measured from saliva assays)”

12/3/2-3. Intriguing statements about optimally combining noisy signals, but explain this more. It won't be obvious to most readers.

We have added some more explanation to this section.

13/1. This is an interesting paragraph where the authors have a chance to discuss what would be most advantageous to the organism. They make the standard argument for perception, but basically punt on having an argument for the pupil.

Indeed, we agree that this point is necessarily speculative, however we think it is interesting for the reader to consider.

13/2/1. "Pupil size affects the ..." is more accurate.

Fixed.

13/2/2 from end. Which "two pathways"? Be clear.

Changed to ‘the pupil and perceptual pathways’